



# Review article: Insuring the green economy against natural hazards - charting research frontiers in vulnerability assessment

Harikesan Baskaran[1], Ioanna Ioannou[1], Tiziana Rossetto[1], Jonas Cels[1], Mathis Joffrain[2], Nicolas Mortegoutte[2], Aurelie Fallon Saint-Lo[2], Catalina Spataru[3]

[1]EPICentre, University College London, London, WC1E 6BT, United Kingdom
[2]AXA Group, 75008 Paris, France
[3]Energy Institute, University College London, London, WC1H 0NN, United Kingdom

*Correspondence to*: Harikesan Baskaran (harikesan.baskaran.21@ucl.ac.uk)

**Abstract.** The insurance of green economy assets against natural hazards is a growing market. This study explores whether currently available published knowledge is adequate for the vulnerability assessment of these assets to natural hazards. A matrix is constructed to demonstrate the vulnerability to functional loss of 37 asset classes in the renewable energy, green construction, resource management, carbon capture and storage, energy storage and sustainable transportation sectors. The 28 hazards adopted range from environmental and geophysical events to oceanic, coastal, and space weather events. A fundamental challenge in constructing the matrix was the lack of an asset-hazard taxonomy for the green economy. Each matrix cell represents the vulnerability of an asset to a specific hazard, based on a comprehensive systematic literature review. A confidence level is assigned to each vulnerability assessment based on a literature density heat map. The latter highlights specific knowledge gaps, and in particular a lack of quantitative vulnerability studies that appropriately represent all functional loss mechanisms in green economy assets. Apart from charting research gaps, a main output of this study is the proposal of a representative asset-hazard taxonomy to guide future, quantitative research that can be applied by the insurance industry.

## 1    Introduction

Climate change is one of the most serious emerging risks faced by the world today. It is partly the consequence of the industrialization of the world economy and its heavy reliance on fossil fuels. In response, governments worldwide have increasingly shifted their focus toward building a greener economy, with many pledging to achieve net-zero greenhouse gas emissions by 2050 (e.g. King's Printer of Acts of Parliament (2019)). This transition toward net zero requires significant investments in assets supporting the green economy. According to McKinsey (2022), global investments in decarbonisation and renewable technologies could reach $800 billion per year by 2030, with corresponding insurance premiums of US$10-15 billion per year. Additionally, green buildings in emerging market cities present a US$24.7 trillion investment opportunity (IFC, 2019), while electric vehicles are projected to account for 35% of the global car market (IEA, 2023).





From an insurance sector perspective, this redirection of exposure (Nature Communications, 2023) towards the green economy offers significant opportunities for insuring new asset types (e.g. Sumaila et al. (2021)). For instance, Buchana and McSharry (2019) estimated an annual probable maximum loss of €1.267 billion from extratropical cyclones, for a portfolio of 38 offshore wind farms in the North Sea. However, this shift introduces challenges, notably in setting insurance premiums for green economy assets against natural hazards. This is particularly critical as insurance plays a pivotal role in enhancing resilience, especially considering that green economy assets are increasingly being established in more hazard-prone regions due to land-use pressures (GCube Underwriting, 2021).

Assessing the vulnerability of green economy assets to natural hazards poses a formidable challenge. While natural catastrophes already account for a fifth of global insurance claims in the construction sector (Allianz Global Corporate & Specialty, 2023), the impact of integrating more green buildings on overall vulnerability and ensuing claims remains uncertain. Similarly, while electric vehicles may be up to 15% more expensive to repair than conventional vehicles (Swiss Re, 2023), their resilience to natural hazards compared to conventional counterparts is unclear. Challenges in vulnerability evaluation arise primarily from the complexity of green economy assets, comprising intricate engineered and/or nature-based systems that are difficult to model. Being relatively new technologies/constructions, they lack historical exposure to extreme climatic or geophysical hazard events. This results in a scarcity of damage and claims data and models for insurers to base their vulnerability and risk evaluations upon. This is true even for the more established solar and wind energy sectors (Lloyd's, 2020), for which some catastrophe modelling and insurance products exist.

Given this data scarcity, insurers could rely more on published knowledge sources to justify their vulnerability models. But, the question arises: how useful is the current literature landscape for this purpose? This study aspires to highlight the limitations of existing published research in informing vulnerability assessments of key green assets to different natural hazards.

In this study, green assets encompass all insurable assets and associated activities that directly contribute to reducing carbon emissions or/and protecting nature. This study focuses on 37 key asset classes, identified across renewable energy, green construction, sustainable transportation, natural resource management, carbon capture/storage and energy storage sectors. Regarding natural hazards, 28 environmental, geophysical, oceanic, and coastal hazards have been selected. A matrix is constructed by the defined assets and hazards, with each intersection representing the vulnerability of the asset to that hazard. Vulnerability, in this context, refers to the likelihood of functional loss and is represented by a qualitative index based on evidence from a systematic review of readily accessible literature for each intersection. Knowledge gaps are highlighted through a literature heat map, and a discussion of these research gaps and their implications for insuring green economy assets against natural hazards is presented.




## 2    Method

The vulnerability assessment of green assets proposed in this study follows a structured approach composed of three steps, as illustrated in Fig. 1. Firstly, 37 assets and 28 hazards are defined, forming a comprehensive list of 1036 asset-hazard intersections, which serve as the basis for our vulnerability matrix. The second step involves a systematic review of global literature relevant to each intersection. This review aims to gauge the relevance and applicability of existing literature in determining vulnerability. In the third step, the results from the literature review are used to assign a qualitative vulnerability

index to each asset-hazard intersection. Additionally, this step involves evaluating the confidence level associated with each vulnerability rating. This evaluation hinges on whether the reviewed literature comprises only qualitative information or includes quantitative fragility/vulnerability functions specific to the analysed asset-hazard pair. In what follows, the three steps are described in greater detail.

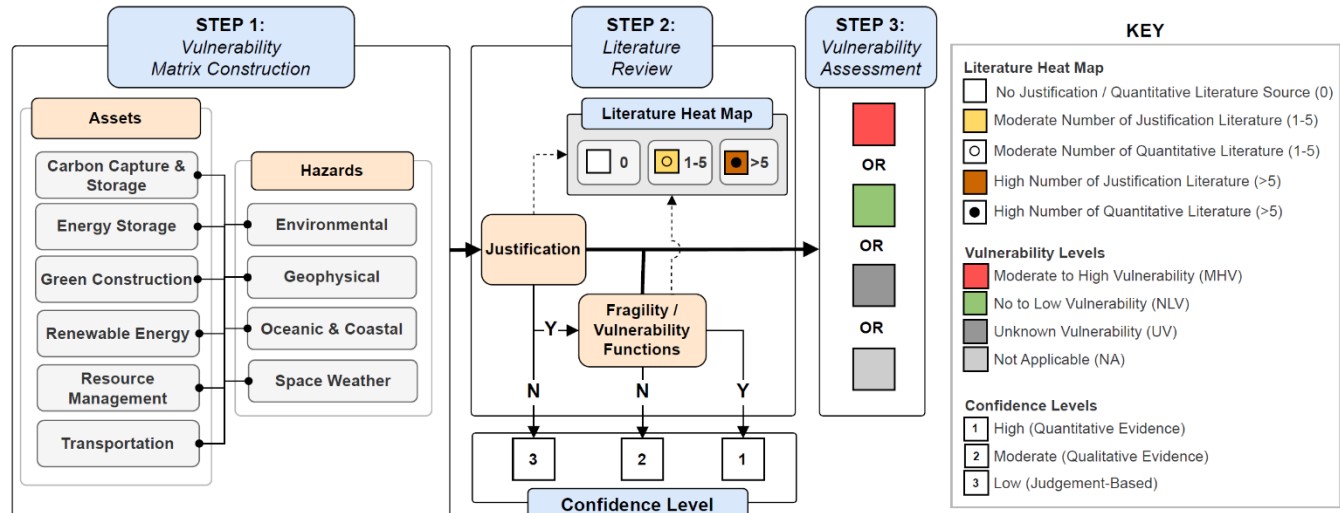


**Figure 1: Flowchart of developed methodology.**

### 2.1    Step 1: Vulnerability matrix construction

It has been observed that existing exposure taxonomy and hazard classification systems do not perfectly align with the needs of vulnerability assessments of green economy assets across various hazard types. Therefore, Step 1 focuses on adapting

existing taxonomy and classification systems to allow the construction of the vulnerability matrix.

#### 2.1.1    Defining green assets

An exposure taxonomy is essential for conducting vulnerability assessments of green assets across a spectrum of natural hazards. However, existing exposure taxonomies are often tailored to specific hazards and cover a limited range of assets. For instance, the Syner-G (Vienna Consulting Engineers, 2014) taxonomy addresses critical infrastructure exposed to



earthquakes, whilst others, like GED4GEM by Gamba et al. (2012), consider multiple hazards but overlook green economy
assets such as green buildings.

Given the lack of a comprehensive exposure taxonomy suitable for this study, we propose a broad definition of exposure
based on categories of green economy assets. This approach aligns with the qualitative and macroscopic nature of
vulnerability assessment conducted here. The assets are intentionally assessed as systems, without a detailed study of all
their interacting components. Overall, six primary sectors of the green economy are considered, drawing from the UK's net-
zero policy (UK Government, 2021): renewable energy sources, green construction, transport with a focus on electric
vehicles, resource management, $CO_2$ reduction, and energy storage. Sub-sector assets are selected to reflect key areas
requiring innovation and investment, which are necessary to achieve the policy's objectives (UK Government, 2021). A total
of 37 assets are identified, with each described (see Table A1).

The majority of assets (28 in total) pertain to renewable energy production, including technologies such as wind, solar,
marine, geothermal energy and hydropower. This includes both established (e.g. offshore bed-fixed wind farms) and
emerging infrastructure and technologies (e.g. floating offshore wind farms, floating photovoltaics, wave energy and tidal
stations). Energy distribution networks connected to renewable energy assets are excluded from the taxonomy. For energy
storage infrastructure, existing assets such as electric batteries and hydrogen storage facilities are considered, as well as
flywheels, which are still at the experimental stage. For biomass and biofuels, the forests and crops respectively, that provide
the raw material, as well as the industrial facilities that turn raw materials into energy, are identified as assets. Facilities for
biogas production are treated separately, reflecting the distinct processes involved in anaerobic digestion and gas
valorisation.

It should be noted that facilities for the production of nuclear energy, although part of the net-zero policy, are not considered
here as this type of energy was deemed clean but not green energy. For the transport sector, both land-based and water-based
electric vehicles are included. The construction sector is represented by only two asset groups, based on green buildings, i.e.
mass timber buildings and ordinary buildings retrofitted to reduce their environmental impact. For $CO_2$ reduction, apart from
the industrial facilities for carbon capture or storage, key ecosystems which capture and store carbon are considered, such as
marshlands and peatlands, as well as protected and unprotected forests. In resource management, three marine ecosystems
are included. Amongst them are coral reefs (The Nature Conservancy, 2022) and mangrove forests (Beck et al., 2020), which
have been the recent focus of the insurance industry, with policies to protect and restore them so they can be effective in
defending coastal areas from natural hazards.





### 2.1.2    Defining hazards

A classification system is needed for representing key characteristics of hazards that can impact asset vulnerability. Many hazard classification systems exist (e.g. UNDRR and ISC (2020); UNDRR (2023)). Similar to in the case of exposure (Sect. 2.1.1), existing hazard taxonomies are defined with mainly engineered assets in mind. They largely ignore hazard

characteristics that can affect the vulnerability of nature-based assets. Notably, hazard duration, identified as a significant factor affecting nature-based asset vulnerability in the literature review conducted during Step 2, is frequently overlooked in existing hazard taxonomies. For example, the vulnerability of crops to a short-term heatwave versus a prolonged rise in temperature due to climate change, can vary significantly. This difference arises because crop yield loss depends on the duration of exposure to temperatures above those optimal for growth (Brás et al., 2021; Zampieri et al., 2017). A new hazard

taxonomy is therefore proposed herein, that is based on the existing hazard taxonomy by UNDRR and ISC (2020), but includes hazard process duration in hazard descriptions. Unlike some risk evaluation taxonomies (e.g. RDLS (2023)), the proposed taxonomy focuses on single hazard processes and ignores triggered, concurrent, sequential or cascading hazards. Unlike RDLS (2023), no intensity measures are defined in the proposed taxonomy, as these are not needed for the vulnerability matrix construction herein.


The proposed hazard classification system consists of 28 natural hazard processes, categorized into four clusters: environmental (16 processes), geophysical (7 processes), oceanic and coastal (4 processes) and space weather (1 process). The considered natural hazards (Table B1), provides detailed descriptions of each hazard.

### 2.2    Step 2: Literature review

A global, systematic review of the literature was conducted for each intersection with the aim to provide justification for the vulnerability rating. The literature reviewed includes five different sources of diminishing perceived quality (Table 1). The literature regarded to be of highest quality (Tier 1) and reliability includes recent peer-reviewed articles or published research reports on the asset-hazard interaction, including fragility/vulnerability functions for key components of the asset. Tier 2 literature comprise peer-reviewed articles and scientific reports from reputable agencies, but do not contain significant or

useful fragility/vulnerability functions for asset components. In the absence of abundant Tier 2 literature, the review focuses on post-disaster reconnaissance reports and/or damage/loss databases from utility providers. Such sources are regarded as of lower quality although still of very good reliability (Tier 3). These are often not peer-reviewed and present observations of asset performance specific to a particular hazard event or asset type, (i.e. are less generalisable). Tier 4 sources comprise design guidelines or manuals for different asset types. Use of these assumes that assets comply with these codes and

guidelines, and hence they were only considered in cases where other literature found did not suggest non-compliance as being common practice. When long duration hazards are assessed, such as rising sea levels, these codes are assumed to be stagnant unless the trajectory of change in design is clear in other literature reviewed. If the literature sources were not





accessible, a broad internet search was conducted to identify news reports or blogs that could provide examples of catastrophic failures of a particular asset due to a given hazard (Tier 5).


Given the variability in the quality and reliability of the literature sources, confidence levels are assigned to each of the asset vulnerability assessments of Step 3, as shown in Table 1. High confidence, CL1, is assigned where the literature review yielded fragility/vulnerability functions for the asset or its key components i.e. where Tier 1 literature was available and deemed reliable and relevant. Moderate confidence, CL2, is assigned to vulnerability assessments made based on Tier 2 to 4

literature sources. Finally, a low confidence (CL3) is assigned to vulnerability assessments made based on Tier 5 sources, or where no relevant literature was found. In the latter case, the vulnerability assessment is made based on the authors' judgement and experience, and from considering the vulnerability of similar asset types. In particular, where there was a lack of literature: e-fuels and hydrogen (small scale) storage tanks were approximated to have failure mechanisms similar to steel storage tanks and silos; hydrogen (large scale) and industrial carbon storage were compared with salt caverns and depleted

oil and gas wells; geothermal, solar CSP power plants and biomass industrial facilities were compared with thermal power plant components, such as cooling systems, fans and turbine generators; land and water-based electric vehicles were compared against fossil-fuel based vehicles where the failure mechanisms were similar.

**Table 1: Systematic literature review with five quality levels and three confidence levels.**

| Source of Literature | Confidence levels (CL) | | |
|---|---|---|---|
| | **CL1:** Quantitative Evidence | **CL2:** Qualitative Evidence | **CL3:** Judgement-based |
| **Tier 1: Peer-reviewed literature including fragility/vulnerability functions for key asset components.** | X | | |
| **Tier 2: Some peer-reviewed literature supplemented with published research reports.** | | X | |
| **Tier 3: Reconnaissance reports, damage datasets from utility providers.** | | X | |
| **Tier 4: Design guidelines and manuals.** | | X | |
| **Tier 5: Internet sources e.g. news reports and company websites.** | | | X |
| **No literature** | | | X |






In the discussion of the vulnerability matrix (see Sect. 3), the high and moderate confidence ratings are grouped together, and are distinguished from the intersections based on low confidence ratings for clarity.

## 2.3 Step 3: Vulnerability assessment

A qualitative vulnerability level is assigned to each asset-hazard pair within the matrix, using the vulnerability scale outlined
in Table 2. These vulnerability levels are structured to mimic a vulnerability function, where asset functional losses are represented as a function of the hazard intensity. Hence, we define Moderate-to-High vulnerability (termed hereafter MHV) as the expectation of significant asset functional loss under low-to-medium hazard intensities and No-to-Low vulnerability (termed hereafter NLV) as the expectation of limited or no asset functional loss under high hazard intensity levels.

During the matrix construction, vulnerability levels are assigned to each asset-hazard pair based on findings from the literature review, or through expert judgement when literature is lacking, as described in Sect. 2.2. In cases where the authors are unable to reasonably assess the likelihood of an assets' functional loss for a specific hazard, a vulnerability level of 'Unknown' (UV) is assigned. For example, a UV level was assigned to biogas (anaerobic digester) industrial facilities under extreme cold events as literature addressing their performance reduction under sub-optimal temperatures (Alvarez and Lidén,
2008) and during the winter season (Pham et al., 2014). Finally, a vulnerability level of 'Not Applicable' (NA) is applied where no literature and/or knowledge exists on the performance of an asset under a specific hazard. Commonly, this is due to the location of the asset being such that it is highly unlikely or never exposed to that specific hazard.

**Table 2: Qualitative vulnerability levels used for this paper.**

| Vulnerability Level | | Description |
|---|---|---|
| **Moderate-to-High** | **MHV** | Significant functional loss expected for a low-to-medium hazard intensity. |
| **No-to-Low Vulnerability** | **NLV** | Limited/No functional loss expected due to a high intensity hazard event |
| **Unknown** | **UV** | No/unreliable information available on the vulnerability of the asset to the specific hazard, but the asset could be exposed to the hazard. |
| **Not Applicable** | **NA** | No information available on the vulnerability of the asset to the hazard, but it is highly unlikely that the asset could be exposed to the hazard. |


The only exception in applying these levels was the assessment of the green retrofitted buildings, where vulnerability was assessed relative to a building without retrofitting, as more (>V), equally (=V) or less vulnerable (<V). For the purpose of the results, >V and =V were assumed to correspond with MHV, whilst <V corresponded with NLV.





It should be noted that for assets comprising multiple components, system-wide losses were considered in the vulnerability assessment. For example, the extent of loss was assessed on the whole forest rather than on individual trees, and for entire wind farms rather than individual wind turbines. However, the criticality of individual components to system functionality is also considered. For example, if a hazard event is likely to destroy only a single tree or wind turbine, the overall impact on the system is limited – hence a limited vulnerability. But if a transformer fails within a wind farm or power plant, this could

lead to a system-level shutdown – hence a high vulnerability.

## 3    Results

The vulnerability ratings of the 1064 unique intersections, conducted according to the proposed methodology, are presented in Table 3. Among these, 396 intersections received MHV ratings, while 341 intersections were classified as NLV. Additionally, 123 intersections were assigned UV ratings due to gaps in the literature.

Of the total assessments, only 337 were based on relevant literature and thus associated with confidence levels (CL) 1 or 2, indicating a higher degree of confidence. In contrast, the majority (526) of judgements were made in the absence of relevant literature (CL3). Moreover 173 intersections were categorized as NA due to improbable geographical coexistence of the asset and hazard. For example, offshore wind farms are not exposed to snow avalanches or glacial melt, warranting an NA vulnerability rating.


The presence of UV ratings highlight a significant gap in the literature. For instance, the absence of literature on the performance of flywheels during sandstorms resolved in a UV rating, reflecting the potential for geographical coexistence. However, for a small number of intersections a UV rating was assigned, despite the presence of literature as it was found to be insufficient, inconclusive or under scientific debate. For example, a conclusive generic vulnerability assessment of

marshlands impacted by rising sea levels cannot be made from published literature, as it is site-specific, and strongly depends on the interaction between tidal imports, the species of vegetation, and depositional processes (Reed, 1995). Similarly, the net impact of storm surge on marshland is under scientific debate. Following Hurricanes Katrina and Rita in 2005, there was a net deposition (5.18 cm on average) of organic and inorganic material, but storm waves can also lead to surface excavation or edge erosion (FitzGerald and Hughes, 2019). Whilst peatlands rapidly decline when impacted by ash

deposits (Zhang et al., 2022), there is insufficient literature identifying the root cause of this relationship, with acidic loading considered a possible reason (Giles et al., 1999).

Table 4 presents a heatmap of the literature for the intersections. In this table, the colour indicates the amount of literature found to justify the vulnerability rating for the asset-hazard intersection. The filled and empty circles represent whether





literature sources containing quantitative expressions of the asset's vulnerability to the hazard exist. Quantitative studies were found for only 4% of the total number of intersections corresponding to 18 assets.

The intersections with the most significant concentration of quantitative studies are associated with the vulnerability of wind farms (both offshore and onshore) to extreme winds and earthquakes. Nonetheless, a closer look to these studies highlights a bias towards the study of specific components. Studies overwhelmingly focus on the vulnerability of the wind turbine towers (42 sources, e.g. Zhang et al. (2023)) and significantly fewer studies focus on the substructure or foundations (17 sources, e.g. Ngo et al. (2023)), blades (10 sources, e.g. Seo et al. (2022)) or nacelles (10 sources, e.g. Rashid and Sarkar (2022)). Commonly, the fragility of the blades and nacelle are interpolated from the fragility of the tower. For the nacelle, only three sources (Cheng et al., 2023; Dueñas-Osorio and Basu, 2008; Hemmati et al., 2019) directly and empirically derive the fragility of acceleration-sensitive nacelle components, such as the generator, inverter and control system.

The next intersection with the highest number of quantitative studies (18 sources e.g. Schwanghart et al. (2018)) is dominated by the seismic fragility assessment of concrete dams. Interestingly, this is the only intersection of hydropower assets with reservoirs where quantitative studies are present, despite the increasing exposure of these assets to other hazards. For example, the proportion of hydropower dams in basins with the highest levels of flood risk is projected to increase by nearly twenty times (e.g., from 2% to 36% of dams) between 2020 and 2050 (Opperman et al., 2022). However, literature for both pluvial and riverine flooding is scarce, with a single, qualitative study (Opperman et al., 2022) assessing the impact of flooding in general to dams, highlighting the importance of ageing.

Biofuel crops are the assets with the highest number of intersections with at least one quantitative study. This literature mainly focuses on the vulnerability of crops to flooding (i.e. riverine, pluvial flooding and storm surge), drought, volcanic ash and extreme wind. Most studies assess the vulnerability of agricultural crops, which could potentially be used for biofuel production, such as Kang et al. (Kang et al., 2016) who estimated yield losses in rice and peanut crops at different flood depths or Craig et al. (Craig et al., 2021) who estimated the fragility of cereal crops to volcanic ash. Only a few studies, have assessed the vulnerability of agricultural crops in general without specifying crop type (e.g. Wang et al. (2022)). Interestingly, no quantitative studies were found for the vulnerability of crops to hailstorms specifically, despite this intersection having one of the oldest insurance products (Randalls and Kneale, 2021).

Overall, the general lack of plentiful quantitative vulnerability studies about some or all the components of the examined assets exposed to most hazards, highlights the need for expert evaluation of vulnerability from the interpretation of other available literature sources.



**Table 3: Asset-hazard vulnerability matrix (with ratings formatted as per Table 2) with confidence levels (as per Table 1). Refer to Tables A1 and B1 for asset and hazard codes.**

| Asset | E_CL | E_DR | E_DSS | E_FRZ | E_GM | E_HLS | E_HTW | E_LTN | E_PFL | E_RFL | E_SA | E_STV | E_TCY | E_WF | E_WS | E_XTC | G_ASF | G_EQ | G_GS | G_LHR | G_LIQ | G_LS | G_PDC | O_OFW | O_SR | O_STS | O_TS | S_SL |
|---|---|---|---|---|---|---|---|---|---|---|---|---|---|---|---|---|---|---|---|---|---|---|---|---|---|---|---|---|
| C_CIC | 3 | 3 | 3 | 3 |  | 3 | 3 | 3 | 3 | 3 | 3 | 3 | 3 | 3 | 3 | 3 | 3 | 3 | 3 | 3 | 3 | 3 | 3 |  |  | 3 | 3 | 3 |
| C_CIS | 3 | 3 | 3 | 3 |  | 3 | 3 | 3 | 3 | 3 | 3 | 3 | 3 | 3 | 3 | 3 | 3 | 2 | 3 | 3 | 3 | 3 | 3 |  |  | 3 | 3 | 2 |
| C_CMA | 2 | 3 | 3 | 3 |  | 3 | 3 | 3 | 3 | 3 |  | 3 | 3 | 3 | 3 | 3 | 3 | 3 |  | 3 | 3 | 3 | 3 |  | 2 | 2 | 3 | 3 |
| C_CPE | 2 | 2 | 3 | 3 |  | 3 | 3 | 3 | 3 | 3 |  | 3 | 3 | 2 | 3 | 3 | 2 | 3 |  | 3 | 3 | 3 | 3 |  | 2 | 2 | 2 | 3 |
| C_CPR | 2 | 2 |  | 3 | 3 | 3 | 3 | 3 | 3 | 3 | 2 | 3 | 2 | 3 | 3 | 2 | 2 | 2 | 3 | 2 | 3 | 3 | 3 |  | 2 |  | 3 | 3 |
| C_CUP | 2 | 2 |  | 3 | 3 | 3 | 3 | 3 | 3 | 3 | 2 | 3 | 2 | 3 | 3 | 2 | 2 | 2 | 3 | 2 | 3 | 3 | 3 |  | 2 |  | 3 | 3 |
| CN_GM | 3 | 3 | 3 | 2 |  | 2 | 3 | 2 | 1 | 3 | 3 | 3 | 2 | 2 | 3 | 1 | 1 | 2 | 3 | 3 | 2 | 3 | 3 |  |  | 3 | 2 | 3 |
| CN_GR | 3 | 3 | 3 | 3 |  |  | 3 | 3 | 3 | 3 | 3 | 3 | 3 | 3 | 3 | 3 | 3 | 3 | 3 | 3 | 3 | 3 | 3 |  |  | 3 | 3 | 3 |
| E_BAT | 3 |  | 3 | 2 |  | 3 | 2 | 3 | 2 | 2 | 3 | 3 | 1 | 2 | 3 | 2 | 3 | 2 | 3 | 3 | 3 | 3 | 2 |  |  | 2 | 2 | 3 |
| E_FLW | 3 | 3 | 3 | 2 |  | 3 | 2 | 3 | 2 | 2 | 3 | 3 | 3 | 3 | 3 | 3 | 2 | 3 | 3 | 3 | 3 | 3 | 3 |  |  | 2 | 3 | 3 |
| E_HYL | 3 | 3 | 3 | 2 | 3 | 3 | 3 | 3 | 2 | 3 | 3 | 3 | 3 | 3 | 3 | 3 | 2 | 3 | 3 | 3 | 3 | 3 | 3 |  |  | 3 | 3 | 2 |
| E_HYS | 3 | 3 | 3 | 2 | 3 | 3 | 2 | 2 | 2 | 3 | 3 | 3 | 2 | 3 | 3 | 2 | 1 | 2 | 3 | 1 | 3 | 3 | 3 |  |  | 2 | 1 | 3 |
| R_BFC | 2 | 1 | 2 | 2 | 2 | 2 | 2 | 3 | 1 | 1 | 2 | 2 | 1 | 2 | 2 | 2 | 1 | 2 |  | 2 | 3 | 2 | 3 |  | 2 | 1 | 2 | 3 |
| R_BFI | 3 | 3 | 3 | 3 | 3 | 3 | 3 | 3 | 1 | 2 | 3 | 3 | 3 | 3 | 3 | 3 | 2 | 3 | 3 | 2 | 3 | 3 | 3 |  |  | 2 | 1 | 3 |
| R_BGA | 3 | 2 | 3 | 2 | 3 | 3 | 2 | 2 | 2 | 2 | 3 | 3 | 2 | 3 | 2 | 2 | 3 | 3 | 3 | 3 | 2 | 3 | 3 |  |  | 2 | 2 | 3 |
| R_BGV | 3 | 3 | 3 | 3 | 3 | 3 | 3 | 3 | 2 | 2 | 3 | 3 | 3 | 3 | 3 | 3 | 3 | 3 | 3 | 3 | 2 | 3 | 3 |  |  | 2 | 2 | 3 |
| R_BMI | 3 | 3 | 3 | 3 | 3 | 3 | 2 | 3 | 2 | 2 | 3 | 3 | 3 | 3 | 3 | 3 | 3 | 3 | 3 | 2 | 3 | 3 | 3 |  |  | 2 | 2 | 2 |
| R_BMW | 2 | 2 |  | 2 | 3 | 2 | 3 | 3 | 2 | 2 | 2 | 3 | 2 | 2 | 2 | 2 | 1 | 2 | 3 | 2 | 3 | 2 | 3 |  | 2 |  | 2 | 3 |
| R_EFU | 3 | 3 | 3 | 3 |  | 3 | 3 | 3 | 3 | 3 | 3 | 3 | 2 | 3 | 2 | 2 | 1 | 2 | 3 | 1 | 3 | 3 | 3 |  |  | 2 | 1 | 3 |
| R_GEO | 2 | 2 | 3 | 2 | 3 | 2 | 2 | 2 | 2 | 2 | 3 | 3 | 1 | 3 | 3 | 2 | 1 | 2 | 2 | 3 | 1 | 2 | 3 |  |  | 2 | 2 | 2 |
| R_HPE | 2 | 2 | 3 | 2 | 3 | 3 | 2 | 2 | 2 | 2 | 2 | 3 | 3 | 2 | 3 | 2 | 2 | 1 | 2 | 3 | 2 | 2 | 3 |  |  |  |  | 2 |
| R_HPR | 2 | 2 | 3 | 2 | 3 | 3 | 3 | 3 | 2 | 2 | 2 | 3 | 3 | 3 | 3 | 3 | 2 | 3 | 3 | 3 | 3 | 2 | 3 |  |  |  |  | 2 |
| R_OCC | 3 | 3 | 3 | 3 |  | 3 | 3 | 3 |  |  |  | 3 | 2 |  | 3 | 3 | 3 | 3 | 3 |  | 3 |  |  | 2 | 2 | 2 | 3 | 2 |
| R_OCR | 3 | 3 | 3 | 3 |  | 3 | 3 | 3 |  |  |  | 3 | 3 |  | 3 | 3 | 3 | 3 | 3 |  | 3 |  |  | 3 | 2 | 3 | 3 | 2 |
| R_OCW | 2 |  | 3 | 3 |  | 3 | 3 | 3 |  |  |  | 3 | 3 |  | 3 | 3 | 3 | 3 | 3 |  |  |  |  | 2 | 3 | 2 | 2 | 2 |
| R_OFB | 3 | 3 | 3 | 2 |  | 1 | 3 | 2 |  |  |  | 3 | 1 | 3 | 2 | 1 | 2 | 1 | 3 | 3 | 2 |  | 3 | 2 |  | 2 | 2 | 2 |
| R_OFF | 3 |  | 3 | 3 |  | 3 | 3 | 3 |  |  |  | 3 | 1 |  | 3 | 2 | 2 | 2 | 3 |  | 2 |  |  | 2 | 3 |  | 2 | 2 |
| R_ONW | 3 | 3 | 2 | 2 | 3 | 1 | 3 | 3 | 2 | 2 | 3 | 3 | 1 | 2 | 2 | 1 | 2 | 1 | 2 | 3 | 3 | 2 | 2 |  | 2 | 3 | 2 | 2 |
| R_SOF | 3 | 3 | 2 | 3 |  | 2 | 2 | 2 |  | 3 |  | 3 | 2 | 2 | 2 | 2 | 2 | 3 | 3 |  | 3 |  |  | 2 |  | 3 | 3 | 2 |
| R_SOL | 2 | 2 | 2 | 3 |  |  | 2 | 3 | 3 | 3 |  | 3 | 3 | 3 |  |  | 3 | 3 | 3 | 3 | 3 | 3 | 3 |  |  | 3 | 3 | 2 |
| R_SOP | 3 | 3 | 2 | 3 |  | 2 | 2 | 2 | 2 | 2 | 3 | 3 | 1 | 2 | 2 | 2 | 2 | 3 | 3 | 3 | 3 | 3 | 3 |  |  | 3 | 3 | 2 |
| R_SOR | 3 | 3 | 2 | 3 |  | 2 | 2 | 2 | 3 | 3 | 3 | 3 | 1 | 2 | 2 | 2 | 2 |  | 3 |  |  |  | 3 |  |  |  |  | 3 |
| RM_PC | 2 |  | 2 |  |  |  | 2 | 3 |  | 2 |  | 1 | 1 | 2 |  |  | 2 | 2 |  | 3 | 3 | 3 | 2 |  | 2 | 2 | 2 | 3 |
| RM_PM | 2 | 2 | 2 |  |  | 2 | 3 | 3 | 3 | 2 |  | 3 | 2 | 2 |  |  | 3 | 3 |  | 2 | 3 | 3 | 3 |  | 2 | 2 | 1 | 3 |
| RM_PS | 1 | 2 | 3 | 2 |  |  | 3 | 3 |  | 2 |  | 3 | 1 | 3 |  | 2 | 3 |  |  | 3 | 2 |  |  |  | 1 | 3 | 2 | 3 |
| T_EVL | 3 | 3 | 3 | 2 |  | 3 | 2 | 1 | 2 | 2 | 3 | 3 | 2 | 2 | 3 | 3 | 2 | 3 | 3 | 3 | 2 | 3 | 2 |  |  | 2 | 2 | 2 |
| T_EVW | 3 | 3 | 3 | 3 |  | 3 | 3 | 3 |  |  |  | 3 | 3 |  |  | 3 | 2 | 3 |  |  |  |  |  | 2 |  | 3 | 2 | 3 |




**Table 4: Literature heat map (mapped as per Fig. 1). Refer to Tables A1 and B1 for asset and hazard codes.**

| Asset | E_CL | E_DR | E_DSS | E_FRZ | E_GM | E_HLS | E_HTW | E_LTN | E_PFL | E_RFL | E_SA | E_STV | E_TCY | E_WF | E_WS | E_XTC | G_ASF | G_EQ | G_GS | G_LHR | G_LIQ | G_LS | G_PDC | O_OFW | O_SR | O_STS | O_TS | S_SL |
|---|---|---|---|---|---|---|---|---|---|---|---|---|---|---|---|---|---|---|---|---|---|---|---|---|---|---|---|---|
| C_CIC | | | | | | | | | | | | | | | | | | | | | | | | | | | | |
| C_CIS | | | | | | | | | | | | | | | | | | ■ | | | | | | | | | | ■ |
| C_CMA | ■ | | | | | | | | | | | | | | | | | | | | | | | | ■ | ■ | | |
| C_CPE | ■ | ■ | | | | ■ | | | ■ | ■ | | | | ■ | | | | | | | | | | | ■ | ■ | ■ | |
| C_CPR | ■ | ■ | | | | | | | | | ■ | | ■ | | | | ■ | | | | ■ | | | | | | | |
| C_CUP | ■ | ■ | | | | | | | | | ■ | | ■ | | | | | | | | | | | | | | | |
| CN_GM | | | | ■ | | ■ | | ■ | ○ | | | | ■ | ■ | | ○ | ● | | | | ○ | | | | | | ■ | |
| CN_GR | | | | | | | | | | | | | | | | | | | | | | | | | | | | |
| E_BAT | | | | ■ | | ■ | ■ | ■ | ■ | ■ | | | ○ | | | | | | | ■ | | | | | ■ | ■ | ■ | |
| E_FLW | | | | ■ | | ■ | ■ | ■ | ■ | ■ | | | | | | | ■ | | | | | | | | ■ | ■ | | |
| E_HYL | | | | | ■ | | | | | | | | | | | | | | | | | | | | | | | ■ |
| E_HYS | | | | ■ | | | ■ | ■ | | | | | ■ | | | ■ | | ○ | ■ | | ○ | | | | | | ○ | |
| R_BFC | ■ | ○ | ■ | ■ | ■ | ■ | | | ○ | ○ | ■ | | ○ | | | | ○ | | | | | | | | ■ | | ○ | |
| R_BFI | | | ■ | | | | | | ○ | | | | | | | | ○ | | | | | | | | | ■ | ○ | |
| R_BGA | | ■ | | ■ | | ■ | ■ | ■ | | | | | ■ | | ■ | | | | | | ■ | | | | | ■ | ■ | |
| R_BGV | | ■ | | ■ | | ■ | ■ | ■ | | | | | ■ | | ■ | | | | | | ■ | | | | | ■ | ■ | |
| R_BMI | | ■ | | ■ | | ■ | | ■ | | | | | ■ | | | | | ■ | | | ■ | | | | | ■ | ■ | |
| R_BMW | ■ | ■ | ■ | ■ | | ■ | | | | | ■ | | ■ | ■ | | ○ | | ■ | | ■ | | | | ■ | | ■ | | |
| R_EFU | | | | | | | | | | | | | | | | | | ○ | | | ○ | | | | | | ○ | |
| R_GEO | ■ | ■ | ■ | ■ | | ■ | ■ | ■ | | | | | ○ | | | ■ | | ○ | | | ○ | | | ■ | ■ | ■ | ■ | |
| R_HPE | ■ | ■dark | | ■ | | ■ | | ■ | | | ■ | | | ■ | | | ● | | | | ■ | | | | | | | |
| R_HPR | ■ | | | ■ | | ■ | | | | | | | | | | | | | | | | | | | | | | |
| R_OCC | | | | | | | | | | | | | ■ | | | | | | | | | | | ■ | ■ | | ■ | |
| R_OCR | | | | | | | | | | | | | | | | | | | | | | | | ■ | | | ■ | |
| R_OCW | ■ | | | | | | | | | | | | | | | | | | | | | | | ■ | | | ■ | |
| R_OFB | ■ | | | ■dark | | ○ | | ■dark | | | | | ● | | ● | | ● | | | | ■dark | | | | | ■ | ■ | |
| R_OFF | | | | | | | | ■dark | | | | | ○ | | | | | | | | | | | | | | ■ | |
| R_ONW | ■ | | ■ | ■dark | | ○ | | ■dark | ■ | | | | ● | ■ | | ○ | | ● | | | | | | ■ | | ■ | ■ | |
| R_SOF | | | | ■ | | | ■ | | | | | | | | | | | | | | | | | ■ | | | ■ | |
| R_SOL | ■ | | | ■ | | | ■ | | | | | | | | | | | | | | | | | ■ | | | ■ | |
| R_SOP | | | | | | | | | | | | | ○ | | | | | | | | | | | | | | ■ | |
| R_SOR | | | | | | | | | | | | | ○ | | | | | | | | | | | | | | ■ | |
| RM_PC | ■dark | | ■ | | | ■ | | | | ■ | | ○ | ○ | | | | | | | | | | | ■ | ■ | ■ | ■dark | |
| RM_PM | ■ | ■dark | | | ■ | | | | | | | | ■dark | | | | | | | ■ | | | | | | | ○ | |
| RM_PS | ○ | | | ■ | | | | | | | | | ○ | | | | | | | | | | | | ■ | ○ | ■ | |
| T_EVL | | | | ■ | | | ■ | ○ | | | | | ■ | | | | | | | ■ | | ■ | | | ■ | ■ | ■ | |
| T_EVW | | | | | | | | | | | | | | | | | | | | | | | ■ | | | | ■ | |





Given the large number of intersections, identifying trends in Tables 3 and 4 can be challenging. Therefore, assets are grouped into 6 sub-classes of green assets and a summary of their vulnerability ratings are provided in Table 5. Each asset is classified according to whether it is natural or engineered, with further subdivisions for natural assets into terrestrial and marine categories, and engineered assets into established and emerging technologies, both onshore and offshore. In each subclass, only the intersections with hazards likely to affect its assets are considered. For example, the offshore wind farms

are not exposed to snow avalanches or glacial melt.

A significant proportion (44%) of natural asset vulnerability ratings were made with high confidence (CL1-2), compared to only 34% for established and emerging engineered assets. With ecosystems, terrestrial assets tend to have a higher vulnerability to natural hazards (50% assigned MHV) compared to marine-based assets (35% assigned MHV). These

evaluations are based on a significant amount of literature. For the engineered assets, the subclasses with the highest vulnerability (MHV), can be ranked as follows: onshore established (50%), onshore emerging (33%), offshore emerging (27%) and offshore established (24%). Emerging onshore engineered assets are also associated with the highest number of UVn ratings (31%).

Overall, the aggregated results highlight the general absence of relevant significant literature for the evaluation of natural hazard vulnerability of green economy assets, and counter-intuitively highlight the scarcity of such literature for established (offshore and onshore) technologies.



**Table 5: Summary of intersections and of overall ratings for asset type.**

| Criteria | Asset | | | | | |
|---|---|---|---|---|---|---|
| | *Natural* | | *Engineered* | | | |
| | | | *Established* | | *Emerging* | |
| | *Terrestrial* | *Marine* | *Onshore* | *Offshore* | *Onshore* | *Offshore* |
| **Total assets** | 6 | 5 | 20 | 4 | 2 | 5 |
| **Total intersections** | 162 | 105 | 520 | 72 | 52 | 90 |
| **For vulnerability ratings: MHV, NLV and UV** | | | | | | |
| Total ratings | *149* | *92* | *495* | *62* | *45* | *84* |
| | CL1-2 | CL3 | CL1-2 | CL3 | CL1-2 | CL3 | CL1-2 | CL3 | CL1-2 | CL3 | CL1-2 | CL3 |
| % of MHV ratings | 35% | 15% | 24% | 4% | 26% | 24% | 19% | 5% | 20% | 7% | 21% | 12% |
| % of NLV ratings | 8% | 30% | 14% | 35% | 9% | 27% | 13% | 45% | 16% | 27% | 12% | 36% |
| % of UV ratings | 2% | 9% | 3% | 18% | 2% | 11% | 0% | 18% | 0% | 31% | 0% | 19% |


Figures 2 - 4 depict the top-five hazards that dominate the MHV ratings for the assets in each subclass. The number of asset ratings is presented by subclass and differentiated by the green economy sector, with the proportion of these ratings assigned a confidence CL1-2 also being provided. It is highlighted that, for these dominant hazards, there is a significant amount of literature allowing high confidence in the vulnerability rating assignment.


It is observed that the dominant hazards vary across the considered sub-classes, and that there is no single hazard that is dominant for all considered engineered and natural green assets. Volcanic hazards significantly affect the vulnerability of most of the terrestrial ecosystems and engineered assets. Pyroclastic flows and lahar appear to be dominant for the vulnerability of established onshore assets and terrestrial ecosystems. By contrast, volcanic ash appears to affect the vulnerability of most emerging onshore and established offshore assets. Coastal hazards appear to affect most asset sub-classes. Tsunami are found to be a major hazard for most terrestrial ecosystems and established assets. Coastal flooding also dominates the vulnerability of both marine ecosystems and onshore emerging assets. Severe storm events are shown to affect most marine ecosystems and all engineered classes except for the onshore established ones.



Some hazards appear to be significant only for assets in each class or subclass. For example, unseasonal patterns are found to affect most natural assets, while solar storms and lightning severely affect all engineered classes with the exception of the onshore established asset types. Landslides and avalanches severely affect only onshore established, engineered assets, while ground shaking and extreme cold severely affect only established offshore assets.

In what follows, a more in depth discussion is provided on the assignment of vulnerability ratings to green economy assets for four key hazards, namely: volcanic and coastal hazards, severe storm and space weather events, as well as low temperature hazards.

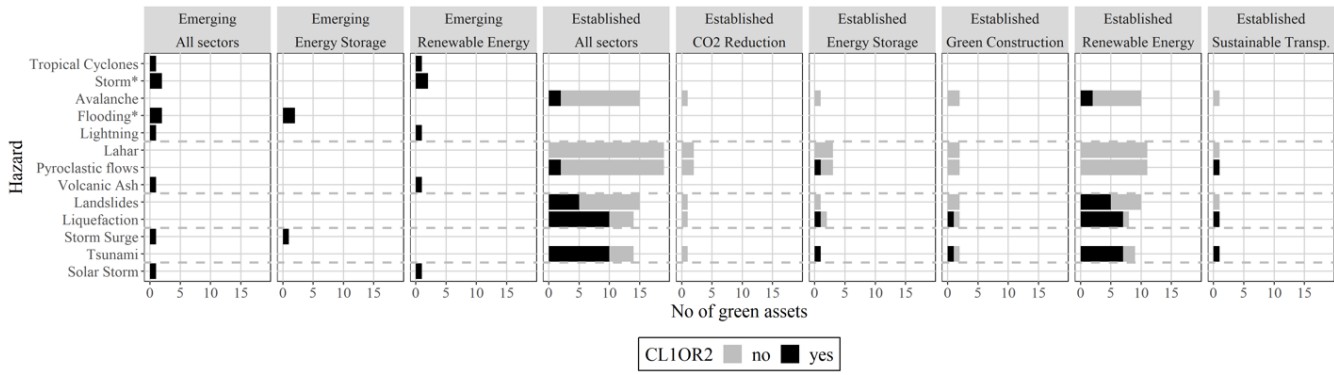

**Figure 2: Top hazards with MHV: Engineered assets (onshore) separated by the green economy sector. *Flooding is a merger of riverine and pluvial flooding, whilst storm is a merger for ice & snow and hailstorm.**

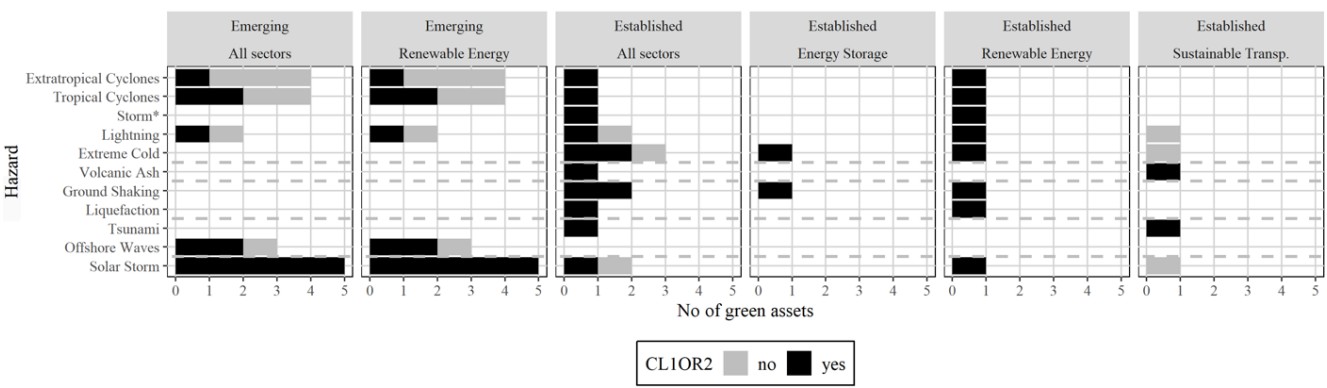

**Figure 3: Top hazards with MHV: Engineered assets (offshore) separated by the green economy sector. *Storm is a merger for**
**winter storm and hailstorm.**





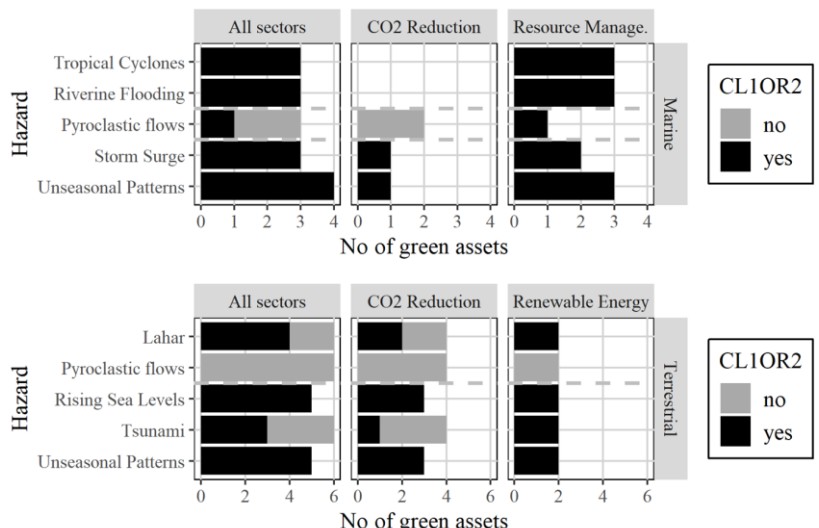

**Figure 4: Top hazards with MHV: Natural assets (marine and terrestrial) separated by green economy sector.**

### 3.1 Discussion of green asset vulnerability to four significant hazards groups

#### 3.1.1 Volcanic hazards

Among the three volcanic hazards assessed, hot, fast-moving pyroclastic flows and lahars are deemed the most mechanically destructive to infrastructure along their direct line of flow (Wardman et al., 2012b). Due to their destructive nature and geographical limitations, construction is typically prohibited in areas prone to these hazards. This lack of exposure results in a scarcity of damage observations for terrestrial engineered assets and of research in this field. Nevertheless, if they were impacted, most terrestrial assets would be highly vulnerable to these hazards and are therefore assigned MHV. Field damage

assessments are available for some nature-based assets, such as forests (e.g. De Guzman (2005)), mangroves (e.g. Joson et al. (2021)) and crops (e.g. Dibyosaputro et al. (2015)), allowing a stronger grounding for their vulnerability evaluation.

Airborne and falling ash ranks as the most geographically widespread volcanic hazard (Wardman et al., 2012b), which explains its top hazard ranking amongst two engineered asset types (established offshore and emerging onshore) supported

by a robust literature base ($\geq$ 50% CL1-2). For example, observations of ash-induced damage on electricity infrastructure (necessary for electrified transportation) from substation flashover, and on geothermal power station operation due to the abrasion of steam condenser fans, have been observed in Japan (Nagai and Nakada, 2022) and Guatemala (Wardman et al., 2012a) respectively. Sea water-based cooling systems for water-based vehicles, are also susceptible to clogging by vesiculated ash, known as 'pumic rafts', which can lead to the overheating of onboard machinery (U.S. Geological Survey,

330 2015).





### 3.1.2 Severe storm events

Severe storms events are found amongst the top hazards for most engineered asset types, particularly those that are offshore and within the renewable energy sector. In most cases, the literature base for its MHV ratings is strong (≥ 50% CL1-2).

Amongst all assets exposed to tropical and extratropical cyclones, offshore (bed-fixed) wind turbines have the highest number of quantitative studies (see Table 4). Despite this, assigning vulnerability to these assets is challenging. To ensure financial feasibility, wind turbines are designed with site-specific wind speeds and turbulence intensities, striking a balance between resistance to extreme wind conditions and optimal power output under normal operation (Larsén et al., 2022). Modern wind turbines are certified against the International Electrotechnical Commission (IEC) 61400-1 (IEC, 2019) design

classes. However, existing vulnerability functions (e.g. Buchana and McSharry (2019)) are not based on turbine designs that are representative of this standard.

An interesting observation was made for floating photovoltaics (FPV) systems with regards to design standards. Hailstorm can cause the fracturing of the glass plate covering most PV modules, resulting in direct damage to the underlying PV

material, or long-term chemical or physical degradation of the internal components due to environmental exposure (Patt et al., 2013). According to the international qualification test IEC 61215-2:2021 (IEC, 2021), all terrestrial FPV panels operating in open-air climates, must withstand a minimum of 11 impacts from 25 mm hailstone at 23 m/s. However, considering that the average maximum hail diameter in Europe is 50 mm (Patt et al., 2013), compliance with international standards does not necessarily imply low vulnerability.

### 3.1.3 Oceanic and coastal hazards

Except for the energy storage sector, at least one oceanic or coastal hazard is amongst the top hazards with MHV ratings. This is reflected across all engineered and natural asset types. Generally, the literature base for their vulnerability ratings is relevant and significant (≥ 50% CL1-2).

The influence of extreme wave conditions produced by tropical or extratropical storms are considered in some of the literature for offshore floating and bed-fixed wind turbines (e.g. Utsunomiya et al. (2013)). Floating wind turbines are seen to be more vulnerable to extreme wave conditions than bed-fixed wind turbines, as slight movements of the floating platform can lead to dramatic vibrations induced by the nacelle and moving blades (Li et al., 2022).

In terms of natural assets, Lagomasino et al. (2021) show that a combination of storm surge and poor drainage caused the highest dieback on record for mangroves in Florida, following Hurricane Irma in 2017. Meanwhile, literature at the





intersection of tsunami and crops used for biofuel, focuses more broadly on production constraints. For example the loss of farming equipment due to the 2009 Samoa tsunami exacerbated the loss of crops in affected areas (FAO, 2009).

### 3.1.4    Space weather and low temperature hazards

Solar storms are the top MHV hazard for most engineered assets in the renewable energy sector. This hazard has the potential to disturb high voltage electrical systems (e.g. transformers) (Radasky, 2011) and control systems (e.g. satellite communication or railway signalling) (Marusek, 2007). Despite being well supported by literature (>50% CL1-2), the number of literature sources for vulnerability evaluation is limited.

With a similarly strong literature base (>50% CL1-2), a broad range of offshore established, engineered assets (varying from energy storage to the sustainable transportation sectors) are seen to be vulnerable to extreme cold events. Li-ion batteries used in battery energy storage applications, are susceptible to reduced ionic conductivity, increased lithium metal dendrite growth and internal resistance, leading to higher temperature in the presence of a flammable electrolyte (Jeevarajan et al., 2022). Whilst water-based electric vehicles lacked a literature base, an MHV rating is still assigned considering the literature

available for its land-based equivalent. A reduction in Li-ion battery performance under extreme cold conditions can reduce vehicle driving range, which is simultaneously affected by the increased battery power demands to maintain cabin temperature (Steinstraeter et al., 2021).

### 3.2    Research gaps

This paper highlights the need for an asset-hazard taxonomy tailored to the green economy. Such a taxonomy should

adequately encompass both emerging and established technologies, while also covering natural assets, to facilitate the production of vulnerability and risk assessment products. Moreover, it is evident that duration must be explicitly considered in hazard definitions, when assessing the vulnerability of natural assets. There is also a need for a species and location-specific taxonomy for natural assets under sea-level rise and flooding hazards, to ensure environmental conditions are appropriately factored into vulnerability assessments. Whilst this study assigns UV ratings to marshlands under sea level rise

and storm surge, it was also noted that Simas et al. (Simas et al., 2001) show that marshlands in the Targus estuary (Portugal) are only susceptible to sea-level rise in a worst case scenario of 0.86 cm per year. Likewise, there is some evidence to suggest that more saline marshes have greater resistance to erosion under storm surge, due to deep rooting and lower soil shear strengths (FitzGerald and Hughes, 2019).

The vulnerability rating exercise conducted has highlighted a lack of literature available for the evaluation of vulnerability generally across all green economy assets and hazards. There is a clear lack of research that presents quantitatively the likely damage or loss of function of green economy assets under natural hazards.



In the developed vulnerability matrix, there are a significant number of intersections with Unknown Vulnerability (UV) ratings, where there is no literature available for the vulnerability evaluation of significant assets with known exposure to the

considered hazards. Areas of concern include intersections between engineered (established onshore and emerging offshore), renewable energy assets, with the hazards of sand storms, unseasonal weather patterns and volcanic ash. These are seen to be significant hazards with UV ratings, with volcanic ash found consistently in the top hazard rankings of all other asset subclasses.

From an analysis of the literature for the established technologies of wind farms, it is observed that there is a need for quantitative literature on the blades and non-structural nacelle components, such as the gearbox. Failure of both of these components are currently leading causes of wind farm insurance claims (Lloyd's, 2020). Despite offshore wind farm losses being dominated by subsea cable failures (Lloyd's, 2020; Allianz Commercial, 2023), no quantitative literature was found either. But even where literature exists, their practical applicability is limited. This study found that all 81 existing offshore

(bed-fixed) wind turbine fragility functions under tropical cyclone conditions use the bespoke NREL 5MW (Jonkman et al., 2009) reference model, which does not comply with the IEC 61400-1 (IEC, 2019). Moreover, the role of connecting infrastructure, such as substations, or the wider distribution and transmission system, should be considered alongside these assets to understand functionality loss. Their fragilities can significantly contribute to the overall vulnerability of a wind farm. In the case of the two recorded instances of earthquake-related wind turbine damage since 1986, electricity grid failure

induced the greatest losses due to downtime (DNV, 2019; Swiss Re, 2017).

As seen in the case of hydropower plants, engineered assets lack a strong literature base under flood hazards. Yet, as seen by AXA Group, there has been a significant increase in insurance claims in relation to hydropower plants impacted by flash floods, with smaller reservoir dam sizes showing a high vulnerability to these events. Assets with long expected lifetimes are

in critical need of flood-related vulnerability assessments, as the design standards that they were initially designed to may not be relevant, especially in the context of climate change.

Whilst the quantitative literature base for crops used for biofuel was identified to be strong, there is a need for more relevant research which determines functional losses in terms of renewable energy production, rather than agricultural yield loss.






## 4 Conclusions

This paper highlights the critical need for a representative green economy asset-hazard taxonomy, essential for developing quantifiable vulnerability assessments relevant to the insurance sector. The limited exposure data for green economy assets, coupled with increasing hazard intensities due to climate change, has led to a difficulty in establishing credible vulnerability
ratings through existing research.

Future vulnerability assessments must consider realistic asset designs and the interaction interplay of components within green economy systems. It is important to consider the compounding effects of multi-hazard events and the cascading impacts of dependent assets failures. In doing this, factors such as the frequency, intensity and duration of hazard events
must be accounted for. For instance, vulnerability may increase for engineered assets as fatigue accumulates over time, but decrease for natural assets, if the return period allows for regeneration.



## Appendices

### Appendix A: Asset Taxonomy

**Table A1.   Asset taxonomy: code and definitions. Classified by Green Economy Sector (SEC): CO2 Reduction (C), Green Construction (CN), Energy Storage (E), Renewable Energy Sources (R), Natural Resource Management (RM), Sustainable Transportation (ST); and Asset Type (AT): natural (NAT) or engineered (ENG), established (ES) or emerging (EM), and environment (ENV), i.e. onshore (ONS), offshore (OFF), terrestrial (TER) or marine (MAR).**

| Code | Asset | SEC | AT | | | Description |
|---|---|---|---|---|---|---|
| | | | NAT /ENG | ES /EM | ENV | |
| **C_CIC** | Carbon Capture: Industrial | C | ENG | ES | ONS | Industrial carbon capture and storage is a three-step process, involving capturing carbon emissions from coal or gas used for power generation or industrial activity, such as steel or cement making; transporting via pipeline or tankers; and then storing it deep underground (UK Health and Safety Agency, n.d.; AXA Group, 2022). Carbon capture technologies used following the combustion of fossil fuels, are the focus of this asset category (UK Health and Safety Agency, n.d.): 1. Post-combustion: Flue gas passed through liquid reactant to separate $CO_2$. 2. Oxyfuel: Fossil fuel burnt in near-pure oxygen. Flue gas therefore only contains $CO_2$ and steam (which is condensed away through a cooling process). |
| **C_CIS** | Carbon Storage: Industrial | C | ENG | ES | ONS/ OFF | Industrial carbon capture and storage is a three-step process, involving capturing carbon emissions from coal or gas used for power generation or industrial activity, such as steel or cement making; transporting via pipeline or tankers; and then storing it deep underground (AXA Group, 2022; UK Health and Safety Agency, n.d.). This asset category will focus on the underground storage of captured carbon. This can include depleted oil or gas reservoirs, or saline aquifers, and is offshore in the UK (UK Health and Safety Agency, n.d.). |
| **C_CMA** | Carbon Capture and Storage: Marshland | C | NAT | - | TER/ MAR | Marshes are all non-peat forming wetland. Whilst it has a substantial content of organic matter within its surface layers, it does not accumulate at a rate fast enough to cause peat formation (Kellner, 2003). Marsh vegetation can capture and store carbon both above and below ground, within the marsh itself, through photosynthesis. Regular flooding can provide carbon outside the ecosystem boundary to |





| Code | Asset | SEC | AT | | | Description |
|------|-------|-----|-----|-----|-----|-------------|
| | | | NAT /ENG | ES /EM | ENV | |
| | | | | | | marshland, in the form of sediment and organic carbon. Marsh vegetation indirectly enable carbon capture by trapping suspended sediment. Moreover, the anaerobic conditions of marshland allow low decomposition rates, and continual marsh growth (Wollenberg et al., 2018). |
| C_CPE | Carbon Capture and Storage: Peatland | C | NAT | - | TER/ MAR | These are freshwater wetlands composed of accumulated, partially decomposed, organic material (from plant matter), known as peat soil (Alshehri et al., 2020). It is of high porosity, has poor nutrient content, and is acidic (Kellner, 2003). It is formed under anaerobic conditions that occur when the water table is close to the ground surface (Alshehri et al., 2020). Despite covering <3% of global land surface, it stores 20% of the world's soil carbon (carbon is captured by plants through photosynthesis) and 60% of carbon in the atmosphere (partial decomposition prevents carbon release into the atmosphere) (Alshehri et al., 2020; UK Centre for Ecology & Hydrology, n.d.). |
| C_CPR | Carbon Capture and Storage: Protected Forests | C | NAT | - | TER | Existing old-growth forests, and regenerating forests, absorb two gigatonnes of carbon per year, globally. Compared to unprotected plantation forests, protected natural forests store relatively more carbon in the form of living biomass, debris and soils (Waring et al., 2020). Moreover, forest carbon is a function of tree size, density and richness, and is believed to be retained well in protected forests (Måren and Sharma, 2021). |
| C_CUP | Carbon Capture and Storage: Unprotected Forests | C | NAT | - | TER | Conservative estimates suggest that large-scale afforestation (planting on former forest land) and reforestation (planting on lands historically without forests) can remove 40-100 gigatonnes of carbon per year, once maturity is reached (the equivalent of a decade's anthropogenic emissions) (Waring et al., 2020). Nevertheless, unprotected forests can be vulnerable to carbon emissions due to forest degradation and deforestation (Måren and Sharma, 2021). |
| CN_GM | Green Buildings-- Mass timber construction | CN | ENG | ES | ONS | This asset class focuses on buildings with primary structures made from engineered timber/ timber mass, e.g. laminate timbre. The technology has evolved rapidly in recent years. For example, the tallest building to be built out of mass timber is a 18-storey, 84.5 m-high residential, mixed-use building, built in 2019 (Moelven, 2024). |



| Code | Asset | SEC | AT | | ENV | Description |
| | | | NAT /ENG | ES /EM | | |
|------|-------|-----|---------|--------|-----|-------------|
| | | | | | | The advantages of engineered timber are they are much less carbon intensive to produce and transport, and have greater renewability and efficiency compared to other structural systems (Abed et al., 2022). The buildings are assumed to be constructed and designed to higher energy efficiency standards which act to reduce the energy consumption required to heat or cool the buildings. Typical technologies that improve the building's energy performance are insulation, double or triple-glazed windows, greater air tightness of the building envelope and air heat exchange units (London Energy Transformation Initiative (LETI), 2021). The design principles that improve a buildings energy performance are broadly the building's surface area to volume ratio and its orientation to sunlight (Hajtmanek et al., 2023). |
| CN_GR | Green Buildings – Retrofitted buildings | CN | ENG | ES | ONS | Deals with existing buildings that are retrofitted to comply with greater energy use standards. Typical retrofitting measure include; an increase of insulation, greater air tightness of the building envelope, or an improvement to double or triple-glazed windows (London Energy Transformation Initiative (LETI), 2021). The most common retrofitting options are chosen, with vulnerability levels assigned as either more or less vulnerable (instead of high, medium or low) to the initial building. |
| E_BAT | Battery (electrical) | E | ENG | ES | ONS | Battery Energy Storage Systems (BESS) store energy using electro-chemical solutions (AIG Energy Industry Group, 2018). Li-ion batteries are the most popular for grid-scale applications, owing to their good grid energy storage capacity and cycle life (can be charged and discharged multiple times), as well as the rapid reduction in their purchase costs (AIG Energy Industry Group, 2018; Jeevarajan et al., 2022). The system is composed of (Jeevarajan et al., 2022): 1. DC electrochemical cells connected in parallel-series configurations to provide the required battery capacity and voltage. These form one of many modules arranged in racks. 2. The battery management system (BMS) allows for controlling battery charge and discharge regimes. |





| Code | Asset | SEC | AT | | | Description |
|---|---|---|---|---|---|---|
| | | | NAT /ENG | ES /EM | ENV | |
| | | | | | | 3. The thermal management system controlling environmental temperature and humidity (in the form of heating, ventilation and air condition (HVAC)). 4. A DC/AC inverter - converting AC from renewable energy systems to be compatible with DC requirements of the cells - or AC/DC inverter - converting DC currents from the storage systems to supply an AC grid. |
| **E_FLW** | Flywheels | E | ENG | EM | ONS | These can be used as short-term energy storage solutions, for stabilising a power grid which is dependent on renewable energy sources with intermittent supply outputs. They store less energy over a smaller time period, compared to batteries. However, they have higher power outputs, have a longer service life (can undergo millions of discharge cycles), are of a relatively smaller size, and occupy less floor area (Wicki and Hansen, 2017). Key components of the system include: a flywheel rotor storing kinetic energy (generally made of composite or metallic materials, which enable higher rotational velocities and moments of inertia), a minimal-loss bearing system (magnetic bearing preferred), a power converter enabling charging and discharging (usually a motor/generator), and a vacuum encloser to reduce losses (Li and Palazzolo, 2022). |
| **E_HYL** | Hydrogen – Large Scale | E | ENG | ES | ONS/ OFF | Large scale hydrogen energy storage (>5 MW) is a form of chemical energy storage in which electrical power is converted into hydrogen. Hydrogen can be used as fuel for gas turbines (Breeze, 2018). The hydrogen must be stored in underground caverns for large-scale energy storage (Breeze, 2018). Salt caverns used extensively for long-term natural gas storage are the focus of this category. |
| **E_HYS** | Hydrogen – Small Scale | E | ENG | ES | ONS | Small scale hydrogen energy storage (<5 MW) is a form of chemical energy storage in which electrical power is converted into hydrogen. Hydrogen can be used as fuel for piston engines or hydrogen fuel cells, with the latter providing the best efficiency (Breeze, 2018). Steel containers can be used for smaller scale storage (Breeze, 2018). |
| **R_BFC** | Biofuel – Crops | R | NAT | - | TER | Biofuel are commonly produced by edible crops, used for the production of liquid fuels for transportation. The oil, starch or sugar of food crops, grown in arable land, are converted into biofuels, e.g., |





| Code | Asset | SEC | AT | | | Description |
|------|-------|-----|---------|--------|-----|-------------|
| | | | NAT /ENG | ES /EM | ENV | |
| | | | | | | bioethanol and biodiesel. Food crops include: corn, soyabeans, sugarcanes, etc. |
| **R_BFI** | Biofuel – Industrial facilities | R | ENG | ES | ONS | This asset class focuses on the industrial facilities used for the conversion of crops or waste to biofuels. |
| **R_BGA** | Biogas – Industrial facilities – Anaerobic digester | R | ENG | ES | ONS | Biogas is a renewable fuel produced by the breakdown of organic matter such as food scraps and animal waste. This asset class focuses on facilities used to produce biogas from waste. The process includes an anaerobic digester where microorganisms are broken down in the absence of oxygen, in a process called anaerobic digestion. The product is then valorised to produce the desired biogas. |
| **R_BGV** | Biogas – Industrial facilities – Valorisation of biogas | R | ENG | ES | ONS | Biogas valorisation is purification of the low quality biogas composed of multiple constituents to a form which is of higher calorific value (e.g. by removing $CO_2$ from the $CH_4$ mixture) and of greater application (e.g. by removing the corrosive hydrogen sulphide, water vapour and siloxane impurities) (Converti et al., 2009; Kapoor et al., 2020). |
| **R_BMI** | Biomass – Industrial Facilities | R | ENG | ES | ONS | Biomass is a renewable energy source, generated from burning wood, plants and other organic matter, such as manure or household waste. This asset class focuses on industrial facilities used for the combustion of solid biomass and energy production. Transformers are included in the vulnerability assessment. |
| **R_BMW** | Biomass – Wood (Forestry) | R | NAT | - | TER | Biomass is a renewable energy source, generated from burning wood, plants and other organic matter, such as manure or household waste. This asset class focuses on wood cultivated for energy production. |
| **R_EFU** | E-Fuels (Synthetic Fuels) Storage | R | ENG | ES | ONS | Gaseous or liquid fuels synthesised using hydrogen and CO2, produced using sustainable electricity (The Royal Society, 2019). Examples include hydrogen, methane, methanol, dimethyl ether and synthetic diesel. Its lifecycle is composed of the following steps (Hänggi et al., 2019):<br>1. Hydrogen production: Electrolysis of water.<br>2. $CO_2$ production: Separated from atmosphere and other sources.<br>3. Chosen e-fuel produced from chemical synthesis and purification. |



| Code | Asset | SEC | AT | | | Description |
|------|-------|-----|-----|-----|-----|-------------|
| | | | NAT /ENG | ES /EM | ENV | |
| | | | | | | 4. Transportation and storage. <br> 5. Oxidation: In an internal combustion engine or fuel cell, producing water vapour and CO2. <br> Only the storage of E-fuels will be considered here. |
| **R_GEO** | Geothermal | R | ENG | ES | ONS | There are two forms of geothermal power generation systems: flash power plants and binary systems. The former uses geothermal heat (>180ºC) to generate steam which directly powers the turbines, and is the most extensively used globally (Atkins, 2013). The latter is used by approximately 15% of geothermal power plants, where temperatures vary between 74 and 180ºC (Atkins, 2013). Water heated by the geothermal thermal reservoir indirectly vaporises the working fluid, which drives the turbine. This is done through a heat exchanger. Despite its low efficiency (10 - 13%), working fluids composed of an ammonia-water mixture can be used to improve system efficiency, with properties ideal for varied operating temperature - this is often referred to separately as the Kalina cycle (Atkins, 2013). Where literature is not available for geothermal power plants, similar elements of other thermal power plants are assessed, such as steam turbines, cooling towers/ ponds, and fans. Transformers are included in the vulnerability assessment. |
| **R_HPE** | Hydro power – Reservoir | R | ENG | ES | ONS | Hydropower systems transform the potential energy of water retained in a reservoir into electrical energy. Hydropower systems consist of several components including a dam, reservoir, a power plant (that consists of turbines, generators, and a powerhouse), switchyards and transmission towers in addition to supplementary systems such as telecommunication systems (Lin and Adams, 2007). <br> The vulnerability of hydropower plants is a function of the combined vulnerability of each of these components. Hydropower production capacity is contingent on the available water mass in the reservoir. Hydropower energy production is therefore vulnerable to any sudden or long term changes in the volume of water that is supplied to the reservoir. |





| Code | Asset | SEC | AT | | | Description |
| --- | --- | --- | --- | --- | --- | --- |
| | | | NAT /ENG | ES /EM | ENV | |
| **R_HPR** | Hydro power – River-run-off | R | ENG | ES | ONS | River-run-off power plants divert the downward flow of rivers into a channel, pipeline, pressurising pipeline (or penstock), to turn turbines which generate electricity. The technology does not store water and is most effective with considerably fast flowing rivers with steady seasonal waters (UN Climate Technology Centre & Network, n.d.). |
| **R_OCC** | Ocean – tidal energy – tidal current stations | R | ENG | EM | OFF | Tidal current stations are powered by the simultaneous kinetic energy of tidal currents. The European Marine Energy Centre has identified broadly five different types of wave converters which are located either onshore, nearshore or offshore (European Marine Energy Centre Ltd, n.d.): 1. Attenuator is a floating device positioned parallel to the wave and produces energy through the relative motion between two arms. 2. Point absorbers are floating devices that converts the motion of the buoyant top relative to the base, into electricity. 3. Oscillating wave surge converters convert the movement of an oscillating arm (underwater) to electrical energy. 4. Oscillating water columns are partially submerged hollow structures which are open below the water line and enclose a column of air. The waves cause the trapped air to rise and fall through a turbine that generates electricity. 5. Overtopping devices capture water into a storage reservoir which is returned to the sea through a low-head turbine that generates electricity. 6. Submerged pressure differential devices are attached to the seabed and are usually located nearshore. They generate electricity using the alternating pressure differential caused by overhead waves. 7. Bulge wave systems are tubes filled with water that are moored to the seabed (which is heading into the waves). Water is allowed to enter through the stern, and waves along the tube generate pressure differentials that creates a 'bulge', which travels to the end of the tube through a low-head turbine that generates power. |





| Code | Asset | SEC | AT | | | Description |
| | | | NAT /ENG | ES /EM | ENV | |
|---|---|---|---|---|---|---|
| | | | | | | 8.   Rotating mass devices induce a mass to rotate within the device, as the device is moved by the waves. The rotating mass is attached to an electrical generator.<br>Transformers are included in the vulnerability assessment. |
| **R_OCR** | Ocean – tidal energy – tidal range stations | R | ENG | EM | OFF | Tidal current power is generated from the rise and fall of sea and ocean waters. Spring and neap tides have a range of about 4 - 12 m and have a potential energy of 1 - 10 MW/km along the seashore (Khan et al., 2017). Power generation capacity follows predictable terrestrial and celestial patterns. Spring tides (high tides) occur during new and full moons and neap tides (low tides) occur during the waxing or waning of half moons (Khan et al., 2017).<br>There are two main types of tidal power stations, tidal range and tidal current stations. Tidal range stations use a tidal barrage technique whereby a dam with electrical turbines is placed across an estuary or along the coast. Energy is generated by the turbines (that may work two directionally) using the periodic water height difference on either side of the dam (Khan et al., 2017).<br>Transformers are included in the vulnerability assessment. |
| **R_OCW** | Ocean – wave energy | R | ENG | EM | OFF | Wave energy converters (WECs) convert the energy from surface waves to electrical energy (ocean waves have both kinetic and potential energies). Ocean surface waves are generated by wind energy blowing over a body of water. Near seashore these waves are typically in the range of 1.3 - 2.8 m high. Wave energy has a comparatively high energy density (2 - 3 kW/m$^2$) compared to solar parks (0.1 - 0.2 kW/m$^2$) and wind farms (0.4 - 0.6 kW/m$^2$) (Khan et al., 2017). There are several wave technologies which are normally situated offshore, nearshore or onshore. A review of different available technologies is presented by Khan et al. (2017). Transformers are included in the vulnerability assessment. |
| **R_OFB** | Offshore wind – Bed-fixed | R | ENG | ES | OFF | The asset class studied here are horizontal axis wind turbines, which are the most common type of turbine used in offshore farms for large-scale applications (Mathew and Philip, 2012). These turbines are typically either two- or three-bladed (Mathew and Philip, 2012). This category will focus on turbines fixed to the sea-bed. Transformers are |



| Code | Asset | SEC | AT | | | Description |
|------|-------|-----|-----|-----|-----|-------------|
| | | | NAT /ENG | ES /EM | ENV | |
| | | | | | | included in the vulnerability assessment. |
| **R_OFF** | Offshore wind – Floating | R | ENG | EM | OFF | The asset class studied here are horizonal axis wind turbines, which are the most common type of turbine used in offshore farms for large-scale applications (Mathew and Philip, 2012). These turbines are typically either two- or three-bladed (Mathew and Philip, 2012). This category will focus on floating wind turbines. Transformers are included in the vulnerability assessment. |
| **R_ONW** | Onshore wind | R | ENG | ES | ONS | This asset class focusses on horizontal-axis wind turbines. They are the most common turbines used in onshore wind-farms for large-scale applications, and they are the most efficient way to transform wind energy to electrical energy (Mathew and Philip, 2012). These turbines are typically either two- or three-bladed (Mathew and Philip, 2012). Transformers are included in the vulnerability assessment. |
| **R_SOF** | Solar power – Floating Photovoltaics (FPVs) | R | ENG | EM | ONS/ OFF | Floating Photovoltaic (FPV) systems have PV arrays and the DC to AC inverters mounted onto a floating platform. Pontoon-type floats are generally used for large-scale FPV plants, and the panels are fixed at a given tilt angle. For small-scale FPV plants, the inverter can be placed on land close to the array (World Bank Group et al., 2019). Transformers are included in the vulnerability assessment. |
| **R_SOL** | Solar power – land-based CSP farms | R | ENG | ES | ONS | Concentrated solar power (CSP) plants, are composed of mirrors which focus solar radiation on a receiver composed of thermal oil or molten salts. These conduct heat and are either directly used to generate electricity with a steam turbine, or are used as mediums for thermal energy storage. 81% of the market are parabolic troughs, with the rest of the CSPs generally being solar towers. Despite its energy storage capabilities (1900 - 2100 kWh/m$^2$), high level of solar radiation are required to make the plants economically viable (World Bank, 2021). |
| **R_SOP** | Solar power – land-based PV farms | R | ENG | ES | ONS | Photovoltaic (PV) systems convert the electromagnetic solar radiation into DC electricity, and require an DC-AC inverter to convert its outputs to AC electricity for grids and local electric loads. Solar cells are generally composed of crystalline silicon, and can be ground-mounted (Marzouk, 2022). This category focuses on commercial, land-based PV farms of all |





| Code | Asset | SEC | AT | | | Description |
|------|-------|-----|---------|-------|-----|-------------|
| | | | NAT /ENG | ES /EM | ENV | |
| | | | | | | sizes. For large-scale farms (>1 MW), land-based PVs are mounted on open racks, to improve air cooling. These make them more efficient, as compared to roof-mounted installations. Arrays can either be fixed or equipped with solar tracking in the x- and/or y-directions (Marzouk, 2022). Transformers are included in the vulnerability assessment. |
| **R_SOR** | Solar power – Roof PVs | R | ENG | ES | ONS | Photovoltaic (PV) systems convert the electromagnetic solar radiation into DC electricity, and require an DC-AC inverter to convert its outputs to AC electricity for grids and local electric loads. Solar cells are generally composed of crystalline silicon, and can be mounted on the rooftop (Marzouk, 2022). This category focuses on commercial and residential roof-mounted PVs of all sizes. |
| **RM_PC** | Protected Ecosystems Marine – Corals | RM | NAT | - | MAR | Coral reefs consist of colonies of many individual marine invertebrate animals, the corals. Corals are fixed in place and they grow slowly. Although corals can be found in all marine environments, coral reefs are only possible in shallow and warm water in the tropical areas (WWT, n.d.). In general, there are four types of corals reefs (Coral Reef Alliance, n.d.): <br> 1. *Fringing reefs:* These are the most common type. They grow near coastlines of islands and continents, and are separated by narrow, shallow lagoons from the shore. <br> 2. *Barrier reefs:* Like fringing reefs, they grow parallel to the coastline, but are separated by deeper, wider lagoons. They can reach the water surface at their shallowest points, posing a navigation barrier. <br> 3. *Atolls:* These are rings of coral commonly located in the middle of the sea, forming protected lagoons. Usually formed when islands with fringing reefs either submerge into the sea or experience sea level rise. <br> 4. *Patch reefs:* They commonly occur between fringing and barrier reefs. These small, isolated reefs grow up from the open island platform base or continental shelf. <br> Coral reefs are located in areas with stable climatic conditions, yet their populations have substantially declined in the past 50 years (Yu, 2012; De'Ath et al., 2012). Climate change and environmental hazards |





| Code | Asset | SEC | AT | | | Description |
| | | | NAT /ENG | ES /EM | ENV | |
|------|-------|-----|---------|-------|-----|-------------|
| | | | | | | can only partially explain this trend (Wilkinson, 2000). Human activity is also to blame. This includes oil spills (Fragoso Ados Santos et al., 2015), shipping traffic and overfishing (Selkoe et al., 2009), wastewater and urban development along the coast (Sale et al., 2011; Burt, 2014). In protected marine areas, where human activity is restricted, fishing is restricted to protect the seaweed-eating fish, which then reduces harmful seaweed and gives baby coral space to grow (Topor et al., 2019). |
| **RM_PM** | Protected Ecosystems Marine – Mangroves | RM | NAT | - | MAR | Mangrove forests consist of shrubs or trees that grow in coastal saline or brackish water and are mainly found in tropical or subtropical areas (Giri et al., 2011; Friess et al., 2019). They help protect the soil from erosion and mitigate the worse effects of tropical cyclones and tsunamis (Danielsen et al., 2005; Mazda et al., 2005; Takagi et al., 2016). |
| **RM_PS** | Protected Ecosystems Marine – Seagrass | RM | NAT | - | MAR | Seagrass are the only flowering plants which grow in marine environments forming large meadows. They create ecosystems which nurture fish populations, weaken storm surges, etc. (Waycott et al., 2009). |
| **T_EVL** | Vehicles: land-based (including supporting infrastructure) | T | ENG | ES | ONS | Electric vehicles are compromised of battery electric vehicles (BEVs), plug-in hybrid vehicles (PHEV) and fuel cell vehicles (FCEVs). They have a 59 - 62% efficiency (compared to the 17 - 21% efficiency of petrol-based vehicles). They convert electric energy from the grid to power at the wheels, with no exhaust, no pollutant emissions, stronger acceleration and with less maintenance (Adderly et al., 2018). This category also includes electric rail, as well as any supporting infrastructure, such as electric charging stations or overhead lines. |
| **T_EVW** | Vehicles: water-based | T | ENG | ES | OFF | Battery powered commercial shipping provides a no-emission alternative to conventional diesel shipping, with the added advantage of relatively lower operational and maintenance costs. As of March 2019, it is estimated that more than 150 battery-powered ships are in operation with around 20 running on full battery power (Jeong et al., 2022). |






## Appendix B: Hazard Taxonomy

**Table B1. Hazard taxonomy: code and definitions. Classified by Hazard Group (GRP): Environmental (E), Geophysical (G), Oceanic and Coastal (O), and Space Weather (S); and Duration (DUR): Short-Term (ST) and Long-Term (LT).**

| Code | Hazard | GRP | DUR | Description |
|------|--------|-----|-----|-------------|
| **E_CL** | Climate Change / Unseasonal Patterns | E | LT | The long-term change in the average weather of a region or variability in its properties. Industrial and human activities have gradually accelerated this process, which includes the increase in the Earth's mean surface temperature (Santos and Bakhshoodeh, 2021). These can include different effects, specific to certain assets, for example: <ul><li>*Melting permafrost:* melting ground at or below 0ºC for at least 2 consecutive years (Biskaborn et al., 2019). It is estimated that one-third of pan-Arctic infrastructure, and 45% of fields used for fossil fuel extraction in the Russian Arctic are in areas where permafrost thawing will occur. The subsequent ground instability will lead to severe damage to the built environment (Hjort et al., 2018).</li><li>*Wind drought:* a reduction in wind speed due to a lack of surface temperature variability between regions.</li><li>*Oceanic acidification:* carbon dioxide dissolution into oceanic water.</li><li>*Shifting ecosystem and microclimates:* Examples include shifting rainfall patterns and humidity in the Amazon or the increasing frequency in locust swarm attacks on crops.</li></ul> |
| **E_DSS** | Dust or Sand Storm | E | ST | Surface wind erosion of drylands, leading to the raising of large volumes of sand particles (>0.06 mm diameter) into the air (Al-Hemoud et al., 2019). Visibility is reduced to less that 1 km, leading to reduced commercial output. There can also be reduced agricultural output due to crop damage and the death of livestock, and infrastructure damage (Al-Hemoud et al., 2019). |
| **E_FRZ** | Extreme cold (freeze) | E | ST | A sudden fall in temperature within 24 hours to extreme low temperatures (well below average), for an extended period of time due to a weather event where there is the cooling of air or the invasion of very cold air (Zuzak et al., 2021; International Federation of Red Cross and Red Crescent Societies (IFRC), 2022). Can lead to negative impacts on people, crops, properties and services, and can be accompanied or preceded by an ice or snow storm (International Federation of Red Cross and Red Crescent Societies (IFRC), 2022). |
| **E_GM** | Glacial Melt | E | ST | Glacial meltwater during unusually hot and wet weather can lead to pressure build up, such that its hydrostatic pressure within the glacier exceeds the cryostatic pressure which constrains it. It bursts through the ice, and discharges downstream creating a flood wave within minutes, impacting nearby communities (Richardson and Reynolds, 2000). |
| **E_HLS** | Hailstorm | E | ST | This is a sub-peril of severe convective storms, formed when there are strong updrafts, large supercooled liquid water contents, high cloud tops and a sufficiently long storm lifetime for |





| Code | Hazard | GRP | DUR | Description |
|------|--------|-----|-----|-------------|
|  |  |  |  | hail formation. Commonly irregular in shape (although sometimes spherical or conical), hailstones are >5 mm in diameter and have similar densities to solid ice (Punge and Kunz, 2016). It can lead to considerable damage to buildings, crops and automobiles (Punge and Kunz, 2016). |
| **E_HTW** | Extreme Heat: Heatwave | E | ST | A period of at least 3 consecutive days (Brimicombe et al., 2021) (and a maximum of 3 weeks), where the regional temperatures (maximum, mean and minimum) are exceeded during a warm period of the year. Generally these lead to strains on healthcare and critical infrastructure (Brimicombe et al., 2021). On occasion, such weather events can lead to a reduction in wind speed (Jiménez et al., 2011). |
| **E_LTN** | Lightning | E | ST | The cloud-to-cloud or cloud-to-ground discharge of current across a large potential difference (Moyo and Xulu, 2021). Often lightning is a sub-peril of severe convective storms, which also encompass other sub-perils: tornados, hail and flash flooding. Together they can lead to large economic losses. Directly, lightning threatens aviation safety, wind turbines, electrical power utilities and transmission lines. It can also start wildfires (Yair, 2018). |
| **E_DR** | Drought | E | LT | A prolonged shortage of water availability, with a particular focus on a lack of precipitation. Onset and conclusion are difficult to determine as effects accumulate slowly and persist after an apparent end (Bullock et al., 2013). A lack of precipitation compared to average (meteorological drought) can often lead to deficiencies in the hydrological system (hydrological drought). This can have further impacts on agricultural resources and other socioeconomic impacts (e.g. reduced hydropower production) (U.S. National Drought Mitigation Center, 2022). |
| **E_PFL** | Pluvial Flooding | E | ST | Surface water flooding can occur when there is heavy rainfall, and are particularly common in cities where urban drainage systems are overwhelmed (Cloke et al., 2017). |
| **E_RFL** | Riverine Flooding | E | ST | Unusually high rainfall volume/intensity, seasonally strong weather (e.g. the monsoon), or sudden melting of snow can lead to a rise of river levels, followed by the overflow or bursting of the banks, which can eventually result in the inundation of the surrounding floodplain (Cloke et al., 2017). |
| **E_SA** | Snow Avalanche | E | ST | An avalanche is the destabilisation and the subsequent flow of part of the snow cover. Two key types of avalanches exist (Louchet, 2020): 1. Slab avalanches: The failure of a weak layer underlying a slab results in downward displacement of the slab, and an avalanche. 2. Loose snow avalanches: It is the growing destabilization of snow grains triggered by only a few grains. Requires cold fluffy snow with low cohesion, and usually occurs on steep slopes. Can also occur on gentle slopes in wet snow conditions. |



| Code | Hazard | GRP | DUR | Description |
|------|--------|-----|-----|-------------|
| **E_STV** | Sudden temperature variation (short-term) | E | ST | A sudden fluctuation in temperature within hours, leading to the disruption of infrastructure system where only usual temperature conditions have been taken into account (Brockway and Dunn, 2020). |
| **E_TCY** | Extreme winds: Tropical Cyclones | E | ST | A weather system composed of large rotating masses of thunderstorms (and wind speeds greater than or equal to 74 m.p.h.), which form over warm ocean waters between latitudes 30ºN and 30ºS (Met Office, n.d.; Shultz et al., 2014). They are also referred to as hurricanes, typhoons and cyclones (Shultz et al., 2014). Such weather events are usually accompanied by (Met Office, n.d.):<br>• High seas: Wave heights of up to 15 metres due to strong winds. Leads to shipping disruption.<br>• Storm surge: Coastal flooding and damage due to several metres of water surge.<br>• Heavy rain: Extensive flooding inland (can release the equivalent of two billion tonnes of moisture picked up per day).<br>• Tornadoes: Extreme wind damage due to tornado development, as the cyclones hit inland. |
| **E_WF** | Wildfire | E | ST | Fire which initiates and propagates in forests and shrubs, and is unplanned, uncontrolled and involuntary (Tedim and Leone, 2020). Wildfires can be differentiated by size (minimum areas used in remote sensing vary between 10 and 100 Ha), and land-use (agricultural fires are removed from remote sensing) (Artés et al., 2019). For a wildfire to substantiate, it must have environmental conditions which promote combustion, an ignition source, and environmental conditions which support the spread of fire (Tedim and Leone, 2020). Assets can be impacted at the periphery. |
| **E_WS** | Winter storm | E | ST | Winter extratropical cyclones, with a specific focus on precipitation in the form of snow and ice. Heavy snowfall causes power outages, infrastructure damage, travel delays and disruption in commercial activities (Hall and Booth, 2017). |
| **E_XTC** | Extreme winds: Extratropical Cyclones | E | ST | Extratropical cyclones occur at greater than 30º latitude from the equator (Frame et al., 2017). They develop when former tropical cyclones move to higher latitude regions of strong horizontal temperature gradients, where the warm tropical and cold tropical air meet (extratropical transition) (Met Office, n.d.; Frame et al., 2017). Wind speeds start at 10 to 20 m.p.h. and can sometimes exceed 73 m.p.h. (similar to tropical cyclones), with the strongest winds often are far away from the centre (by contrast, for tropical cyclones it is at the centre) (Met Office, n.d.; U.S. National Park Service, n.d.). High winds and precipitation (only rain, and no snow or ice will be considered in this instance) are major hazards associated with tropical cyclones (Frame et al., 2017). In offshore environments, the atypical waves produced will be considered. |





| Code | Hazard | GRP | DUR | Description |
|------|--------|-----|-----|-------------|
| **G_ASF** | Volcanic: Ash Fall | G | ST | Material produced during volcanic eruptions, which is less than 2 mm in diameter, and is the most widely distributed eruption product (Wilson et al., 2012). It affects populations over large areas, and can be detrimental to public health, industry, aviation and critical infrastructure, despite not being as destructive as lahars and pyroclastic flows (Wilson et al., 2012). |
| **G_EQ** | Earthquake & Ground Shaking | G | ST | The sudden release of strain energy in the Earth's crust leading to waves of shaking radiating from the source (the focus) (British Geological Survey, 2022a). Here, only the direct impacts from ground shaking are considered. |
| **G_GS** | Ground Settlement | G | LT | A type of land subsidence, where there is relatively slow, moderate downward vertical displacement of the Earth's surface (as opposed to collapse, which is sudden or catastrophic in nature). This is due to the underground instability, a load superimposed on the surface, or both, due to natural processes or anthropogenic activities leading to instability in the natural environment (Marker, 2013). |
| **G_LHR** | Volcanic: Lahar | G | ST | A mudflow made up of volcanic debris and hot or cold water, moving at 10-100 k.p.h. Heavy rainfall or eruptions which involve meltwater from ice or snow can lead to such flows, which can gather more loose material as they travel down river valleys. Flows with a 60:40 sediment-to-water ratio have a wet concrete consistency, whilst lower ratios lead to less viscous flows resembling torrential flooding (British Geological Survey, 2022b). These only impact specific areas. |
| **G_LIQ** | Earthquake & Ground Shaking: Liquefaction | G | ST | The loss of strength and stiffness of soils during earthquake ground shaking, leading to ground deformation. This leads to building damage at the surface and infrastructure damage, including to underground utilities (Huang and Yu, 2013). |
| **G_LS** | Landslides | G | ST | A mass wasting process on natural or engineered slopes, whereby rock, debris or earth moves down the slope under gravity. Movement can take the form of flowing, sliding, toppling, falling, spreading or a hybrid combination (Gariano and Guzzetti, 2016). Anthropogenic activities, weather events (e.g. snow melting, temperature change or precipitation) and other natural hazards (including earthquakes or volcanic activity), can trigger landsides (Gariano and Guzzetti, 2016). |
| **G_PDC** | Volcanic: Pyroclastic Density Currents | G | ST | Hot (between 100ºC and 600ºC), fast flowing (usually >110 k.p.h.) currents consisting of rock debris and gas, which usually flow along the sides of a volcano to lower ground under gravity (British Geological Survey, 2022b). These only impact specific areas, and pose a fire risk. |
| **O_OFW** | Offshore Waves | O | ST | Strong winds, applied over a large distance and time in the ocean can lead to waves of large amplitudes and wavelengths. During storms, there is a mixture of waves travelling in variable directions and with different properties, creating rough sea conditions. Water |



| Code | Hazard | GRP | DUR | Description |
|---|---|---|---|---|
| | | | | vessels and offshore infrastructure are particularly vulnerable to this hazard (New Zealand National Institute of Water and Atmospheric Research, n.d.). Only atypical waves are considered (whilst wind may trigger this, it will not be considered in the vulnerability assessment). |
| **O_SR** | Rising sea levels | O | LT | A long-term rise in sea level (according to the U.S. National Oceanic and Atmospheric Administration (2022) this is between 22 and 24 cm since 1880) due to two components of global warming (NASA, 2022): <br> 1. Melting ice sheets and glaciers add more water to the sea. <br> 2. Seawater expands as it is warmed. <br> Such a hazard can not only inundate low-lying coastal areas, but can also increase storm surge elevations and inundation distances. Therefore the vulnerability of coastal areas to hazards other than sea-level rise is also higher (FitzGerald et al., 2008). |
| **O_STS** | Coastal Flooding: Storm Surge | O | ST | Abnormal rise in sea level above the typical, astronomical tide levels, usually due to strong winds from a cyclone, forcing water onshore (Australian Bureau of Meteorology, 2022). |
| **O_TS** | Coastal Flooding: Tsunami | O | ST | Waves generated by earthquakes, undersea landslides or volcanic eruptions. They travel larger distances and inundate a larger land area at the coast, compared to storm surges (Australian Bureau of Meteorology, 2022). |
| **S_SL** | Solar Storm | S | ST | Large-scale magnetic eruptions at the Sun, lead to the acceleration of charged particles (predominantly positively-charged protons) to high-velocities. These travel to Earth over millions of km within minutes. The Earth's magnetosphere (which shields the Earth from low-energy charged particles) guides the particles to the North and South poles as it enters the atmosphere. There is a radiation risk to humans from the energised protons, whilst electronic equipment are also susceptible to damage (U.S. National Oceanic and Atmospheric Administration, n.d.). |






**Code availability**

Tables 3, 4 and 5 were generated using Microsoft Excel and Word. Figures 2, 3 and 4 were generated using R and Microsoft Excel. The code and files used to create these figures and tables can be made available upon specific requests made to the corresponding author, and approval by AXA Group.

**Data availability**

The list of literature that has been adopted for the literature heat map can be made available upon specific requests made to the corresponding author, and approval by AXA Group.

**Author contribution**

All authors contributed to reviewing the literature and writing the paper.

**Competing interests**

The authors declare that they have no conflict of interest.

**Acknowledgements**

HB received funding from the UK Engineering and Physical Sciences Research Council (EPSRC) Doctoral Training Partnership (DTP) [Grant Number: EP/W524335/1], with industry sponsorship from AXA Group.



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
