# Peer review of "Review article: Insuring the green economy against natural hazards - charting research frontiers in vulnerability assessment"

_Natural Hazards and Earth System Sciences, 2024_

## Referee Comment (RC1)

Reviewer Comments: Insuring the green economy against natural hazards – charting research frontiers in vulnerability assessment

The paper presents data on which hazards to green infrastructure have been estimated in a way that can be useful to insurers looking to provide coverage. To do this, they identify 37 asset classes and 28 hazards, and show which of the combinations of asset class and hazard have existing research on the topic. This is very useful for research moving forward, as it identifies areas of the green economy where there is little to no research on the possibility of damage to green infrastructure.

The main deficiency of the paper is that there is little insight to how important each intersection is, apart from excluding the cases that are unlikely to occur and so do not have research. Some of the asset classes identified will be, or already are, a major part of the green economy, such as land-based PV farms. The paper makes a start on this question by breaking it down into plausible impacts and implausible impacts. This is a good start, and generally good enough to publish the paper as a way to move the discussion forward. However, it may be useful to address which of the plausible but understudied impacts might be most relevant.

Main points of consideration

1. The paper does not currently distinguish between direct damage to infrastructure versus losses due to reductions in performance. Both are discussed throughout, but it is not made clear how insurance contracts typically work in this context, if they only generally cover direct damage, or if there is usually cover for performance, or if it is context dependent. Additional context would be useful to understand where the main gaps are, and which are most relevant for insurance markets. As an example, it is noted that quantitative studies were found for 4% of the intersections, but it would be useful to have a sense of what share of the value this represents.
2. While insurance is one way to prevent losses from climate-related damage, there may also be adaptation measures that can be taken to reduce or eliminate losses. For example, the authors note that flywheels operating in dust storms have not been researched, however, there may be low cost ways to protect flywheels from the effects of these storms. Other adaptations may be more costly, such as shielding solar PV from hail damage, in which case insurance may be the better way to reduce risks. While the authors do not need to conduct an analysis of the potential for adaptation in each case, there should be some discussion of how this may factor into the analysis of which combinations are plausible. It may also be reasonable to ignore this possibility, but then this assumption should be stated.
3. Relatedly, it is noted that volcanic risks for example are generally avoided by prohibiting development nearby. This would seem to raise some ambiguity as to whether these risks are then "Unknown" or "Not Applicable." It would help to have some guidance on how this ambiguity is resolved.

Minor Comments

4. At the top of page 2, it is highlighted that the expansion of green assets presents an opportunity for insurers, but it could also be noted that the provision of insurance may help to encourage the further expansion of green assets. It may be useful to have an estimate of the total size of the green asset market that may be insurable, to give an overall sense of scale.
5. The rules for the inclusion of assets are not entirely clear. For example, it seems that pumped hydro storage could fit into the category of energy storage infrastructure along with electric batteries and hydrogen, but is not included. This is similar for the exclusion of new buildings that are not mass timber.
6. In Table 1, there is no need for 3 separate columns under CL2. Tiers 2-4 all fall under CL2, they can be in the same column. The additional columns may create confusion.
7. The sentence beginning on line 178 does not appear to be complete, it is difficult to understand what is meant.
8. In Table 3, it is unclear what is meant when a rating of "Unknown" has a confidence level. While most of them are level 3, some are also level 2, indicating there is some qualitative evidence. It should be explained what is meant by this combination.
9. Table 3 may be difficult to read for people with red/green colorblindess.
10. In Table 4, there should be a legend explaining the meaning of the colors that appear on the heatmap. It is also not clearly explained what it means to have no circle, an empty circle, or a filled circle.
11. In Section 3.2, it would be useful to have a specific discussion of how the taxonomy would be used by insurers, or how the lack of a taxonomy is hindering the provision of insurance, rather than simply stating that the paper "highlights the need for an asset-hazard taxonomy tailored to the green economy."
12. The sentence beginning on line 402 is not clear (I assume it is meant that no quantitative evidence for losses due to cable failures was found).
13. In the Conclusion, there could be some discussion on how much insurers generally rely on scientific research to inform their understanding of risks, compared to how much do they have to rely on it because it is a new and emerging sector. Presumably they could conduct their own studies as well, if there is a potential market in a given sector.

---

## Author Comment (AC1)

**Response to Anonymous Referee #1 Comments: Insuring the green economy against natural hazards – charting research frontiers in vulnerability assessment**

Note: Line numbers refer to the original manuscript. Line numbers from the revised manuscript (without tracked changes) are provided in brackets.

R1.1. The paper presents data on which hazards to green infrastructure have been estimated in a way that can be useful to insurers looking to provide coverage. To do this, they identify 37 asset classes and 28 hazards, and show which of the combinations of asset class and hazard have existing research on the topic. This is very useful for research moving forward, as it identifies areas of the green economy where there is little to no research on the possibility of damage to green infrastructure.

**Author Response:** *Thank you for your positive comments. We appreciate your support for this research work as a vehicle for moving research forward on the natural hazard vulnerability of the green economy.*

R1.2. The main deficiency of the paper is that there is little insight to how important each intersection is, apart from excluding the cases that are unlikely to occur and so do not have research. Some of the asset classes identified will be, or already are, a major part of the green economy, such as land-based PV farms. The paper makes a start on this question by breaking it down into plausible impacts and implausible impacts. This is a good start, and generally good enough to publish the paper as a way to move the discussion forward. However, it may be useful to address which of the plausible but understudied impacts might be most relevant.

**Author Response:** *Thank you for highlighting this area for improvement. Amendments have been made to address which of the plausible, but understudied, impacts might be most relevant. Our focus has been around four asset groups that we consider to be the most relevant: natural, hydropower, onshore wind, and established, solar PV assets.*

*An additional sentence with a new reference to Chaplin-Kramer et al. (2022) was added, explaining the need for focussing on natural assets as a means of mitigating climate change, starting line 388 (now line 432, Section 3.2, Research gaps):*

> *"The protection and restoration of forests, wetlands and peatlands could sequester 9 Gt $CO_2$ per year by 2050 (Chaplin-Kramer et al., 2022), highlighting the urgent need for developing affordable, representative insurance products for natural assets to mitigate the impacts of climate change."*

*Two additional sentences (starting line 392 (now lines 439 – 442), Research Gaps) have been added highlighting that established, engineered assets, experiencing rapid growth, in particular solar PV and onshore wind assets (IRENA, 2023), are in more urgent need of quantitative research:*

> *"The rapid growth in established, engineered assets such as utility-scale solar PV and onshore wind technologies, underlines the importance addressing this gap. Both assets led the global deployment of renewable energy technologies in 2022, with nearly 90% of them assessed to be more cost effective than new fossil fuel alternatives (IRENA, 2023)."*

*Starting at line 416 (new line 467, Section 3.2, Research Gaps), given that hydropower assets take up a large share of global renewable energy capacity (IRENA, 2023), a new sentence is added to highlight the importance of focusing on hydropower assets amongst other ageing engineered assets:*

> *"Ageing hydropower assets must be prioritised amongst them, taking the highest (41%) capacity share of global renewable energy sources (IRENA, 2023), the largest green economy sector of this study."*

*A new subsection (4.1, Future work) has been added under Section 4, Conclusion. Subsequently, lines 427 – 431 (new lines 501 – 506) have now been moved under this subsection. Within this subsection, a new paragraph starting after sentence ending line 431 (new lines 508 – 515) summarises the importance of focusing on the key aforementioned asset groups in future work:*

> *"In the short-term, future quantitative assessments by researchers should prioritize established engineered assets that are experiencing rapid market growth, particularly solar PV and onshore wind installations. While these assets incorporate site conditions in their design standards, lower initial investment costs and the decreasing availability of optimal locations, may result in reduced due diligence during the planning stage, shifting greater risk to insurers. Additionally, with the changing climate, it is crucial to prioritise ageing hydropower assets in research efforts due to their extensive global presence and the potential risks associated with their long-term operation. Simultaneously, natural assets urgently require the development of vulnerability models that are tailored to specific locations and species, ensuring their capacity to mitigate the impacts of climate change effectively."*

*The aforementioned addition, in part responds to Anonymous Reviewer #2 (R2.8 and R2.10).*

Main points of consideration

R1.3. The paper does not currently distinguish between direct damage to infrastructure versus losses due to reductions in performance. Both are discussed throughout, but it is not made clear how insurance contracts typically work in this context, if they only generally cover direct damage, or if there is usually cover for performance, or if it is context dependent. Additional context would be useful to understand where the main gaps are, and which are most relevant for insurance markets. As an example, it is noted that quantitative studies were found for 4% of the intersections, but it would be useful to have a sense of what share of the value this represents.

*Author Response: To address this comment we have added a section starting at line 171 (now lines 190 – 195, Section 2.3, Step 3: Vulnerability assessment) with an additional reference to Rose and Huyck (2016), to provide more context:*

> *"Functional loss is a broad term covering both performance reduction and direct damage, commonly referred to as business interruption and property damage respectively in insurance contracts, where they are generally covered. Vulnerability to functional loss in this paper does not specifically distinguish between business interruption and property damage, as the two are closely linked, with business interruption commonly resulting from property damage and the subsequent recovery process (Rose and Huyck, 2016). Consequently, the existing literature rarely provides sufficient detail on the contribution of each to the overall losses observed."*

*We decided to highlight the lack of literature focussing on business interruption or performance-based losses, within the 4% of intersections with quantitative studies. The sentence starting line 220 (now line 249, Section 3, Results) now reads:*

> *"Quantitative studies were found for only 4% of the total number of intersections corresponding to 18 assets, with the vast majority being fragility (or physical vulnerability) functions focussing on property damage."*

*And sentence starting line 391 (now line 437, Section 3.2, Research gaps) now reads:*

> *"There is a clear lack of research that presents quantitatively the likely loss of function of green economy assets under natural hazards, and in particular functions assessing indirect business interruption losses."*

R1.4. While insurance is one way to prevent losses from climate-related damage, there may also be adaptation measures that can be taken to reduce or eliminate losses. For example, the authors note that flywheels operating in dust storms have not been researched, however, there may be low cost ways to protect flywheels from the effects of these storms. Other adaptations may be more costly, such as shielding solar PV from hail damage, in which case insurance may be the better way to reduce risks. While the authors do not need to conduct an analysis of the potential for adaptation in each case, there should be some discussion of how this may factor into the analysis of which combinations are plausible. It may also be reasonable to ignore this possibility, but then this assumption should be stated.

**Author Response:** *Thank you for mentioning the importance of adaptation measures and mitigation technologies. The vulnerability assessment conducted in this study considers a generalised asset that is not site-specific and is in as-built condition. Therefore adaptation and mitigation measures are specifically not considered. This is clarified through an additional statement starting at line 176 (now lines 201 – 203, Section 2.3, Step 3: Vulnerability assessment):*

> *"In this assessment assets are generalised, and adaptation measures or mitigation technologies are not considered, unless literature mentions them as a minimum requirement. Therefore the worst-case vulnerability rating is provided in the matrix."*

*We do however acknowledge that this assumption is an inherent limitation of the study, and address this as an area for future work in the new sentence starting at line 428 (now line 502, Section 4.1, Future work):*

> *"This includes the impact of common mitigation technologies and adaptation measures on assets assessed as high vulnerability."*

R1.5. Relatedly, it is noted that volcanic risks for example are generally avoided by prohibiting development nearby. This would seem to raise some ambiguity as to whether these risks are then "Unknown" or "Not Applicable." It would help to have some guidance on how this ambiguity is resolved.

**Author Response:** *Thank you for highlighting this ambiguity. Our assessment of vulnerability for volcanic (pyroclastic and lahar) hazards does align with the criteria mentioned in Section 2.3, Step*

*3: Vulnerability assessment. Given the limited availability of insurance products for the hazards, the potential for geographical coexistence is implied, and an "NA" rating cannot be applied. A reasonable judgement can also be made using the limited literature available, and literature related to other assets, hence an "Unknown" rating was not given. We acknowledge that this was not clearly explained, and therefore we have made the following amendments in lines 316 - 319 (now lines 351 - 358, Section 3.1.1, Volcanic hazards), which includes new references to DCCA Hawaii Insurance Division (2023) and Aspinall et al. (2011):*

> *"Due to their destructive nature and geographic constraints, areas prone to these hazards generally see limited engineered or industrial construction. This is often enforced through restrictions in private and state insurance coverage, as seen by AXA Group. Consequently, this lack of exposure results in a scarcity of damage observations and research on terrestrial engineered assets in volcanic regions. Despite the limited availability of insurance products for these hazards (e.g. DCCA Hawaii Insurance Division (2023)), this study reasonably assesses vulnerability by drawing on literature related to other assets (e.g. Aspinall et al. (2011)). If impacted, most terrestrial assets would be highly vulnerable to these hazards, thus they are assigned a high vulnerability rating (MHV)."*

**Minor Comments**

R1.6.  At the top of page 2, it is highlighted that the expansion of green assets presents an opportunity for insurers, but it could also be noted that the provision of insurance may help to encourage the further expansion of green assets. It may be useful to have an estimate of the total size of the green asset market that may be insurable, to give an overall sense of scale.

**Author Response:** *Thank you for your comment on this. The sentence starting on line 35 (now line 38) has been amended to note that the provision of insurance encourages the expansion of green asset deployment:*

> *"This is particularly critical as insurance plays a pivotal role in enhancing resilience and the expanding green asset deployment, especially considering that they are increasingly being established in more hazard-prone regions due to land-use pressures (GCube Underwriting, 2021)."*

*The sentence starting on Line 26 gives the size of the green asset market that might be insurable. For clarity and to give a better sense of scale, we can expand on the definition of decarbonisation and renewable energy technologies, specified that insurance premiums are only for capital expenditures, and provided an additional estimate of the green economy's extent from FTSE Russell (2023) as its definition varies with literature source. Lines 26 – 29 (now lines 26 – 32) now read:*

> *"FTSE Russell (2023) estimates the green economy to be the fourth largest standalone sector in the global stock markets, with a US$6.5 trillion value. Additionally, green buildings in emerging market cities present a US$24.7 trillion investment opportunity (IFC, 2019), while electric vehicles are projected to account for 35% of the global car market (IEA, 2023). According to McKinsey (2022), global investments in decarbonisation (e.g. carbon capture and storage, as well as electric vehicle charging) and renewable energy (e.g. solar power, as well as onshore and offshore wind) technologies could reach US$800 billion per year by 2030, which corresponds to insurance premiums of US$10-15 billion per year on capital expenditures alone."*

*The aforementioned addition also responds to Anonymous Reviewer #2 (R2.2).*

R1.7.  The rules for the inclusion of assets are not entirely clear. For example, it seems that pumped hydro storage could fit into the category of energy storage infrastructure along with electric batteries and hydrogen, but is not included. This is similar for the exclusion of new buildings that are not mass timber.

**Author Response:**

*Thank you for your comment regarding the clarity of asset inclusion criteria. We have added further clarity regarding how we developed this initial version of a green asset taxonomy, in the context that there is no operational taxonomy available for the insurance sector.*

*A new sentence added at line 86 (now line 89, Section 2.1.1, Defining green assets) reads:*

> *"At present, there is no operational taxonomy available for the green economy assets that can be used by the insurance sector."*

*Lines 91 – 95 (now lines 95 – 103, Section 2.1.1, Defining green assets) now read:*

> *"Six primary sectors of the green economy are considered, based on the UK's net-zero policy (UK Government, 2021): renewable energy sources, green construction, transport with a focus on electric vehicles, resource management, $CO_2$ reduction, and energy storage. Through discussions between authors at University College London and AXA Group we identified the operational requirement of the insurance sector. This collaboration led to the development of an initial version of an operational green asset taxonomy. It includes at least two common, insurable sub-sector assets per sector, ensuring broad coverage. These assets were chosen to reflect key areas requiring innovation and investment to meet the policy's objectives (UK Government, 2021). Each asset is defined at a level that can be priced and is practical for the insurance industry, and includes assets currently being insured. In total, 37 assets are identified and described in detail (see Table A1)."*

*The aforementioned addition, in part responds to Anonymous Reviewer #2 (R2.4).*

*We agree that this initial version of an asset-hazard taxonomy is in need of expansion and standardisation. We have highlighted that this must be done with inputs from the wider insurance industry, and have added this in a new paragraph inserted before line 427 (new lines 498 – 499, Section 4.1, Future work):*

> *"To support this, the operational asset-hazard taxonomy proposed in this study is in need of expansion and standardisation with inputs from the wider insurance industry."*

*The aforementioned addition, in part responds to Anonymous Reviewer #2 (R2.8).*

R1.8.  In Table 1, there is no need for 3 separate columns under CL2. Tiers 2-4 all fall under CL2, they can be in the same column. The additional columns may create confusion.

**Author Response:** *Thank you for highlighting this. We have merged the three separate columns accordingly.*

R1.9. The sentence beginning on line 178 does not appear to be complete, it is difficult to understand what is meant.

**Author Response:** *Thank you for bringing this to our attention. The sentence on line 178 (now line 205) has now been amended and reads as follows:*

*"For example, a UV level was assigned to biogas (anaerobic digester) industrial facilities under extreme cold events as literature was only found addressing their performance reduction under sub-optimal temperatures (Alvarez and Lidén, 2008) and during the winter season (Pham et al., 2014)."*

R1.10. In Table 3, it is unclear what is meant when a rating of "Unknown" has a confidence level. While most of them are level 3, some are also level 2, indicating there is some qualitative evidence. It should be explained what is meant by this combination.

**Author Response:** *Thank you for your comment. The small number of "Unknown" vulnerability ratings that are made based on existing literature, indicate that the literature was either insufficient, inconclusive or under scientific debate. In these instances, a reasonable assessment the likelihood of an asset's functional loss for a specific hazard, cannot be made. This is explained in the sentence starting in line 208 (now line 237). A few examples are also given in the sentences starting on lines 178, 209, 212, 214 (now lines 205, 238, 241, 243).*

R1.11. Table 3 may be difficult to read for people with red/green colorblindess.

**Author Response:** *Thank you for highlighting this. We wanted the 'red' to perceive a negative connotation, i.e. moderate-to-high vulnerability, and the 'green' to provide a positive connotation, i.e. no-to-low vulnerability. We have provided an magenta/green alternative to ensure that this differentiation is more noticeable for those with red/green colour blindness, whilst still carrying the intended connotations. Please see below.*

*Sample from Table 3 before amendment:*

| Asset | Hazard | | | | |
|---|---|---|---|---|---|
| | E_CL | E_DR | E_DSS | E_FRZ | E_GM |
| C_CIC | 3 | 3 | 3 | 3 | |
| C_CIS | 3 | 3 | 3 | 3 | |
| C_CMA | 2 | 3 | 3 | 3 | |
| C_CPE | 2 | 2 | 3 | 3 | |

*Sample from Table 3 after amendment:*

| Asset | Hazard | | | | |
|-------|------|------|-------|-------|------|
| | E_CL | E_DR | E_DSS | E_FRZ | E_GM |
| C_CIC | 3 | 3 | 3 | 3 | |
| C_CIS | 3 | 3 | 3 | 3 | |
| C_CMA | 2 | 3 | 3 | 3 | |
| C_CPE | 2 | 2 | 3 | 3 | |

*This colour changes have also been reflected in Figure 1 and Table 2.*

R1.12. In Table 4, there should be a legend explaining the meaning of the colors that appear on the heatmap. It is also not clearly explained what it means to have no circle, an empty circle, or a filled circle.

**Author Response:** *The caption for Table 4 (Line 256 (now line 292)) mentions that it is "mapped as per Fig. 1", as the legend for the colour and circles is provided in Figure 1. We agree that wording is insufficient to signpost this to the reader, therefore we have changed the wording of the caption to "mapped and formatted as per Fig. 1".*

R1.13. In Section 3.2, it would be useful to have a specific discussion of how the taxonomy would be used by insurers, or how the lack of a taxonomy is hindering the provision of insurance, rather than simply stating that the paper "highlights the need for an asset-hazard taxonomy tailored to the green economy."

**Author Response:** *Thank you for raising this. We have now added a specific discussion in Section 3.2, after sentence ending line 379 (new lines 418 – 422), to elaborate on the need for an asset-hazard taxonomy for insuring the green economy:*

*"Adopting a standardized classification system across the insurance sector, will enable the interoperability of functional loss data and analyses between researchers and insurers. In scenarios where exposure data is scarce, such a system will accelerate the development of quantitative fragility and vulnerability functions for complex green economy assets, facilitating their direct application within the insurance sector."*

R1.14. The sentence beginning on line 402 is not clear (I assume it is meant that no quantitative evidence for losses due to cable failures was found).

**Author Response:** *Thank you for highlighting the lack of clarity, and your assumption is correct. We have amended the sentence beginning on line 402 (now line 453) to improve clarity:*

*"Despite offshore wind farm losses being dominated by subsea cable failures (Lloyd's, 2020; Allianz Commercial, 2023), no fragility or vulnerability functions were found for cables either. But even where literature exists, their practical applicability is limited."*

R1.15. In the Conclusion, there could be some discussion on how much insurers generally rely on scientific research to inform their understanding of risks, compared to how much do they have to rely on it

because it is a new and emerging sector. Presumably they could conduct their own studies as well, if there is a potential market in a given sector.

**Author Response:** *Thank you for your comments. As suggested we have added an additional discussion, in the conclusion. It is important to note that the insurance industry typically does not rely on scientific research, when sufficient historical loss data is available. We addressed this point in a new addition before line 422 (new lines 482 – 486, Section 4, Conclusion), within the same paragraph:*

> *"Assessing the vulnerability of green assets under natural hazards is a significant challenge for the insurance industry. As an emerging sector, the green economy lacks historical loss data, which traditional insurance models depend on. In such cases, insurers may use scenarios analyses based on a combination of published engineering models and internal expert judgement. In this context, the study assessed the potential of using existing, published literature to support the development of these models."*

**References**

Allianz Commercial: A turning point for offshore wind: Global opportunities and risk trends, https://www.allianz.com/content/dam/onemarketing/azcom/Allianz_com/press/document/Allianz-Commercial_A-turning-point-for-offshore-wind.pdf, last access: 25 September 2023.

Alvarez, R. and Lidén, G.: The effect of temperature variation on biomethanation at high altitude, Bioresour Technol, 99, 7278–7284, https://doi.org/10.1016/j.biortech.2007.12.055, 2008.

Aspinall, W., Auker, M., Hincks, T., Mahony, S., Nadim, F., Pooley, J., Sparks, S., and Syre, E.: GFDRR, Volcano Risk Study - Volcano Hazard and Exposure in GFDRR Priority Countries and Risk Mitigation Measures , 2011.

Chaplin-Kramer, R., Neugarten, R. A., Sharp, R. P., Collins, P. M., Polasky, S., Hole, D., Schuster, R., Strimas-Mackey, M., Mulligan, M., Brandon, C., Diaz, S., Fluet-Chouinard, E., Gorenflo, L. J., Johnson, J. A., Kennedy, C. M., Keys, P. W., Longley-Wood, K., McIntyre, P. B., Noon, M., Pascual, U., Reidy Liermann, C., Roehrdanz, P. R., Schmidt-Traub, G., Shaw, M. R., Spalding, M., Turner, W. R., van Soesbergen, A., and Watson, R. A.: Mapping the planet's critical natural assets, Nat Ecol Evol, 7, 51–61, https://doi.org/10.1038/s41559-022-01934-5, 2022.

DCCA Hawaii Insurance Division: HOMEOWNERS INSURANCE - Town Hall Meeting with Sen. San Buenaventura and Rep. Ilagan, https://cca.hawaii.gov/ins/files/2023/08/Pahoa-Community-8-23-23.pdf (last access: 31 July 2024), 2023.

FTSE Russell: Investing in the green economy 2023, https://www.lseg.com/content/dam/ftse-russell/en_us/documents/research/investing-in-the-green-economy-2023.pdf (last access: 1 August 2024), 2023.

GCube Underwriting: Hail or High Water, 2021.

IEA: Global EV Outlook, Paris, https://iea.blob.core.windows.net/assets/dacf14d2-eabc-498a-8263-9f97fd5dc327/GEVO2023.pdf, last access: 28 July 2023.

IFC: Green Buildings: A Financial and Policy Blueprint for Emerging Markets, Washington, D.C., Green Buildings: A Financial and Policy Blueprint for Emerging Markets (last access: 28 July 2023), 2019.

IRENA: Renewable power generation costs in 2022, Abu Dhabi, https://www.irena.org/Publications/2023/Aug/Renewable-Power-Generation-Costs-in-2022, last access: 1 December 2023.

Lloyd's: Renewable energy risk and reward: Risks and technologies, https://assets.lloyds.com/assets/pdf-renewable-energy-risk-and-reward-renenergy-risksandtechnologies/1/pdf-renewable-energy-risk-and-reward-RenEnergy_RisksandTechnologies.pdf (last access: 4 July 2023), 2020.

McKinsey: Capturing the climate opportunity in insurance, https://www.mckinsey.com/industries/financial-services/our-insights/capturing-the-climate-opportunity-in-insurance (last access: 14 July 2023), 2022.

Pham, C. H., Vu, C. C., Sommer, S. G., and Bruun, S.: Factors Affecting Process Temperature and Biogas Production in Small-scale Rural Biogas Digesters in Winter in Northern Vietnam, Asian-Australas J Anim Sci, 27, 1050–1056, https://doi.org/10.5713/ajas.2013.13534, 2014.

Rose, A. and Huyck, C. K.: Improving Catastrophe Modeling for Business Interruption Insurance Needs, Risk Analysis, 36, 1896–1915, https://doi.org/10.1111/risa.12550, 2016.

UK Government: Net Zero Strategy: Build Back Greener, https://assets.publishing.service.gov.uk/government/uploads/system/uploads/attachment_data/file/1033990/net-zero-strategy-beis.pdf (last access: 14 July 2023), 2021.

---

## Author Comment (AC2)

**Response to Anonymous Referee #2 Comments: Insuring the green economy against natural hazards – charting research frontiers in vulnerability assessment**

Note: Line numbers refer to the original manuscript. Line numbers from the revised manuscript (without tracked changes) are provided in brackets.

R2.1. The study provides a comprehensive overview of the vulnerability of green economy assets to natural hazards, identifying key gaps in the literature and proposing a structured taxonomy for future research. The methodology is well-defined, and the use of a systematic literature review to construct a vulnerability matrix is robust. However, there are areas where the paper could be improved to enhance its clarity, depth, and utility.

**Author Response:** *We thank the reviewer for their positive feedback on the paper, and for summarising areas for improvement. Please find our response to your points for improvement below.*

R2.2. The introduction effectively sets the context for the study. However, it could benefit from a clearer explanation of the significance of the green economy in the context of climate change and insurance (lines 24-35). Including more recent statistics or projections could provide a stronger foundation.

**Author Response:** *Thank you for your comment. We have addressed this in our response to Anonymous Reviewer #1 (R1.6).*

R2.3. While the paper identifies gaps in the literature (lines 50-55), it would be useful to elaborate on why these gaps exist. Are they due to the novelty of the technologies, lack of historical data, or other reasons? This could help guide future research more effectively.

**Author Response:** *Thank you for highlighting this point. We have highlighted the reasons for the lack of literature when we discussing specific assets and hazards. For example, where there is a lack of studies for pyroclastic flows and lahars impacting terrestrial engineered assets, due to the practice of avoiding construction in volcanic areas (see our response to Anonymous Reviewer #1 (R1.5)).*

*However, in response to this comment, amendments were made to lines 156 - 157 (now lines 174 - 177, Section 2.2, Step 2: Literature review), mentioning low exposure history as a reason for literature gaps:*

> *"In the latter case, the vulnerability assessment is made based on the authors' judgement and experience. It was observed by the authors that assets with a low exposure history, including relatively new technologies/constructions, generally lacked an academic literature base. Here, the vulnerability of similar asset types were considered".*

*Please also review our response to R2.7, which shows a lack of literature for mechanical components of the nacelle, due to modelling complexities.*

*To emphasise the key reasons why these gaps exist in published literature, we have made amendments to the conclusion lines 422 – 425 (now lines 486 - 491, Section 4, Conclusion). It now reads:*

> *"The limited exposure data for complex green economy assets, as seen in the insurance sector; the insufficient alignment of published vulnerability assessments with design standards and insurance needs; and the increasing intensity of hazards due to climate change, have all contributed to the difficulty in establishing credible vulnerability ratings through existing research. This paper highlights the critical need for a representative green economy asset-hazard taxonomy, which is essential for guiding researchers in developing quantitative vulnerability assessments that are relevant to the insurance industry."*

R2.4. The proposed taxonomy is central to the study (lines 75-120). It would be beneficial to provide more justification for the selection of specific assets and hazards. For instance, why were certain assets or hazards prioritized over others? This could help readers understand the choices made and the potential limitations.

**Author Response:** *Thank you for suggestion. With regards to asset inclusion criteria, we have addressed this in our response to Anonymous Reviewer #1 (R1.7). In line with the amendments made for asset inclusion, changes have been made to lines 124 – 126 (now lines 132 – 137, Section 2.1.2, Defining hazards), to clarify the hazard inclusion criteria:*

> *"A new, operational hazard taxonomy for the insurance sector is therefore proposed herein, that is based on the existing hazard taxonomy by UNDRR and ISC (2020), with the addition of hazard process duration in hazard descriptions. Similar to the approach used for green assets (Section 2.1.1), hazards were prioritised and consolidated through author discussions. This process ensured that the selected hazards were the most relevant for the chosen assets, and practical for use at an operational level within the insurance industry."*

R2.5. The systematic review process is well-outlined (lines 130-155). However, providing more detail on the search strategy, databases used, and inclusion/exclusion criteria could enhance transparency and replicability.

**Author Response:** *Thank you for your suggestion. We have added more details on search strategy, databases used and inclusion/ exclusion criteria after sentence ending line 136 (new lines 147 – 152, Section 2.2, Step 2: Literature review):*

> *"A combination of asset and hazard names from the developed green economy taxonomy, alongside the terms 'fragility' and 'vulnerability', were searched within easily accessible, web-based literature databases (e.g. Google Scholar). Where literature results were found to be insufficient to give a representative vulnerability rating, alternative keywords were used, before reference lists of relevant published literature were hand searched. All literature found were included in the assessment, and were only excluded when a potential or definitive report or discussion of effect, damage, vulnerability, or loss of function was not present."*

*An example was added to lines 147 – 149 (now lines 165 – 167, Section 2.2, Step 2: Literature review) to clarify the meaning of 'broad internet search':*

> *"If the literature sources were not accessible, a broad internet search (e.g. via Google) was conducted to identify news reports or blogs that could provide examples of catastrophic failures of a particular asset due to a given hazard (Tier 5)."*

R2.6. The presentation of the vulnerability matrix (lines 190-210) is comprehensive. However, it might be helpful to include a few illustrative examples or case studies to demonstrate how the matrix

can be applied in real-world scenarios. This could make the findings more tangible for practitioners.

**Author Response:** *Thank you for your comment. The presented matrix provides qualitative vulnerability ratings at each intersection, that are intended to be used alongside the literature heat map to identify research gaps in existing published literature. These ratings need to be transformed into a quantitative metric in order to be practically applicable by the insurance sector, for example through the macroseismic method used by Lagomarsino and Giovinazzi (2006). Ideally these qualitative ratings must be validated with the independent development of quantitative fragility/vulnerability functions, as mentioned in our response to comment R2.9. The second sentence starting on line 170 (now line 190) has been removed from section 2.3 (Step 3: Vulnerability assessment) to avoid confusion regarding the practical applicability of the vulnerability matrix.*

R2.7. The literature heat map (lines 220-250) is a valuable addition. However, it could be enhanced by a more detailed discussion of the trends observed. For instance, why are certain asset-hazard intersections more researched than others? Are there specific barriers to research in certain areas?

**Author Response:** *Thank you for your insightful feedback. In response to your recommendation, we have expanded the discussion on the trends observed in the literature heat map and included a new reference to EPRI (2021). The following details have been added before the sentence on line 249 (new lines 279 – 284, Section 3, Results):*

> *"As expected, natural assets and established engineered assets, which are predominant in the green economy, show a greater availability of quantitative literature. Despite this, the complexity of certain assets and the associated hazard processes can present significant barriers to research, even when a quantitative literature base exists. For example, structural components of wind turbines, such as the tower, have more fragility functions, because wind and seismic loads can be directly associated with the structure's limit states. In contrast, mechanical components within the nacelle, such as the gearbox, have limited literature on the composition of their multiple subcomponents, and how these experience indirect loading (EPRI, 2021)."*

R2.8. The discussion provides a good overview of the key findings (lines 285-390). However, it could benefit from a more detailed analysis of the implications for different stakeholders, such as policymakers, insurers, and researchers. What specific actions should they take based on these findings?

**Author Response:** *Thank you for your recommendation. We have addressed through the following amendments. A summary is presented in a new paragraph added before line 427 (new lines 493 – 494, Section 4.1, Future work):*

> *"In this section, recommendations for research and insurance industry practitioners are made, with the intention that outputs from these stakeholders will guide policymakers in changing codes of practice for the protection of green economy assets."*

*Actions to be taken by the for the insurance sector based on key findings is addressed in our response to Anonymous Reviewer #1 (R1.7). Please also see our response to R2.10, which elaborates on the implications of having more practically-applicable, quantitative vulnerability models (which this study found a lack of) for the insurance sector.*

*The implications of key findings for researchers is also addressed through our response to Anonymous Review #1 (R1.2) .*

R2.9. While the paper outlines future research needs (lines 380-400), it could be more specific. For example, identifying specific technologies or methodologies that could be used to address the identified gaps would be helpful.

**Author Response:** *Thank you for highlighting more specific research needs. We have provided more guidance on the approaches that can be taken to develop quantitative fragility/ vulnerability functions, with a new reference to Ioannou et al. (2017). A new paragraph is now added after line 419 (new lines 474 – 480, Section 3.2, Research gaps), reading:*

*"To address the identified gaps, quantitative fragility and vulnerability functions can be developed using empirical models, that rely on systematic observations of functional loss and its root causes, as well as a clear understanding of their link. When empirical data is limited, analytical methods may be employed as an alternative, provided numerical modelling is feasible. In cases where both empirical data and numerical modelling are unavailable, expert elicitation approaches can be used, as seen in the case of Ioannou et al. (2017), which quantified the vulnerability of reinforced concrete buildings to various fire intensities. Depending on data availability, a hybrid approach that combines empirical, analytical, and expert-based methods may also be utilised."*

R2.10. The conclusion effectively summarizes the main findings (lines 420-430). However, it could be strengthened by reiterating the practical implications and the urgency of addressing the identified research gaps in light of climate change and increasing natural hazards.

**Author Response:** *Thank you for your comment. We have addressed this in our response to Anonymous Reviewer #1 (R1.2), where we added a discussion on short-term research needs. Three further sentences have also been added in a new paragraph inserted before line 427 (new lines 495 to 498, Section 4.1, Future work), highlight the importance and urgency of addressing research gaps:*

*"For the insurance sector specifically, practically-applicable, quantitative vulnerability models are needed to reduce uncertainty in pricing insurance premiums. More broadly, such models will improve the industry's internal risk monitoring efforts, and help confidently achieve regulatory requirements. In turn, this will help de-risk investment in the growing green economy and improve its resilience to natural hazards."*

**References**

EPRI: Wind Turbine Gearbox Reliability Assessment: Value of Increased Reliability and Reduced Operations and Maintenance Costs, https://www.epri.com/research/products/000000003002021422 (last access: 1 August 2024), 2021.

Ioannou, I., Aspinall, W., Rush, D., Bisby, L., and Rossetto, T.: Expert judgment-based fragility assessment of reinforced concrete buildings exposed to fire, Reliab Eng Syst Saf, 167, 105–127, https://doi.org/10.1016/j.ress.2017.05.011, 2017.

Lagomarsino, S. and Giovinazzi, S.: Macroseismic and mechanical models for the vulnerability and damage assessment of current buildings, Bulletin of Earthquake Engineering, 4, 415–443, https://doi.org/10.1007/s10518-006-9024-z, 2006.

UNDRR and ISC: Hazard definition and classification review (Technical Report), https://www.undrr.org/publication/hazard-definition-and-classification-review-technical-report (last access: 4 August 2023), 2020.

---

## Author Response (AR2)

**Author's response: Insuring the green economy against natural hazards – charting research frontiers in vulnerability assessment**

Dear Editor,

Thank you for your decision to accept our manuscript with corrections, and for inviting us to submit files for the production process.

Please find below a detailed point-by-point response to requested technical corrections in your report:

E1. References: I have noticed that in some places, you have double brackets in referencing. For example, (e.g. Sumaila et al. (2021)). Can you please make sure to correct these throughout the manuscript and in line with journal guidelines?

*Author Response: We thank the Editor for bringing this to our attention. Accordingly we have made amendments throughout the manuscript, so that it in-line with journal guidelines. The in-text citation(s) on:*

1. *Line 25 of the accepted manuscript (now line 25) now reads:*
   o *"(e.g. King's Printer of Acts of Parliament, 2019)".*

2. *Line 35 of the accepted manuscript (now line 35) now reads:*
   o *"(e.g. Sumaila et al., 2021)".*

3. *Line 126 of the accepted manuscript (now line 128) now read:*
   o *"(e.g. UNDRR and ISC, 2020; UNDRR, 2023)".*

4. *Line 137 of the accepted manuscript (now line 139) now reads:*
   o *"(e.g. RDLS, 2023)".*

5. *Line 256 of the accepted manuscript (now line 258) now reads:*
   o *"(42 sources, e.g. Zhang et al., 2023)".*

6. *Lines 256 to 257 of the accepted manuscript (now line 258 to 259) now reads:*
   o *"(17 sources, e.g. Ngo et al., 2023)".*

7. *Line 257 of the accepted manuscript (now line 259) now reads:*
   o *"(10 sources, e.g. Seo et al., 2022)".*

8. *Line 257 of the accepted manuscript (now line 259) now reads:*
   o *"(10 sources, e.g. Rashid and Sarkar, 2022)".*

9. *Line 262 of the accepted manuscript (now line 264) now reads:*
   o *"(18 sources, e.g. Schwanghart et al., 2018)".*

10. *Line 275 of the accepted manuscript (now line 277) now reads:*
    o *"(e.g. Wang et al., 2022)".*

11. *Line 355 of the accepted manuscript (now line 357) now reads:*

- "(e.g. DCCA Hawaii Insurance Division, 2023)".

12. *Line 356 of the accepted manuscript (now line 358) now reads:*
    - "(e.g. Aspinall et al., 2011)".

13. *Lines 358 to 359 of the accepted manuscript (now line 360) now reads:*
    - "(e.g. De Guzman, 2005)".

14. *Line 359 of the accepted manuscript (now line 361) now reads:*
    - "(e.g. Joson et al., 2021)".

15. *Line 359 of the accepted manuscript (now line 361) now reads:*
    - "(e.g. Dibyosaputro et al., 2015)".

16. *Line 379 of the accepted manuscript (now line 381) now reads:*
    - "(e.g. Buchana and McSharry, 2019)".

17. *Line 395 of the accepted manuscript (now line 397) now reads:*
    - "(e.g. Utsunomiya et al., 2013)".

E2. Captions: I suggest that the caption for Figure 1 could be more descriptive and self-explanatory.

**Author Response:** *We thank the Editor for highlighting this. Following amendments line 79 of the accepted of the accepted manuscript (now lines 79 to 81), it now reads:*

*"Figure 1: Flowchart of developed methodology. An asset-hazard taxonomy is defined for vulnerability matrix construction in Step 1. At each matrix intersection, a systematic literature review is conducted and a heat map is developed in Step 2, enabling confidence levels to be assigned to the vulnerabilities ratings made in Step 3."*

We hope these corrections are sufficient to initiate the preparation of the final version of our manuscript.

Regards,
Harikesan Baskaran and co-authors.